# Final-Model-Only Data Attribution
# with a Unifying View of Gradient-Based Methods

**Dennis Wei**
IBM Research
dwei@us.ibm.com

**Inkit Padhi**
IBM Research

**Soumya Ghosh**
Merck Research Labs

**Amit Dhurandhar**
IBM Research

**Karthikeyan Natesan Ramamurthy**
IBM Research

**Maria Chang**
IBM Research

## Abstract

Training data attribution (TDA) is concerned with understanding model behavior in terms of the training data. This paper draws attention to the common setting where one has access only to the final trained model, and not the training algorithm or intermediate information from training. We reframe the problem in this "final-model-only" setting as one of measuring sensitivity of the model to training instances. To operationalize this reframing, we propose *further training*, with appropriate adjustment and averaging, as a gold standard method to measure sensitivity. We then unify existing gradient-based methods for TDA by showing that they all approximate the further training gold standard in different ways. We investigate empirically the quality of these gradient-based approximations to further training, for tabular, image, and text datasets and models. We find that the approximation quality of first-order methods is sometimes high but decays with the amount of further training. In contrast, the approximations given by influence function methods are more stable but surprisingly lower in quality.

## 1 Introduction

Training data attribution (TDA, or sometimes simply "data attribution") refers to the attribution or explanation of ML model behavior in terms of its training data. Existing methods for TDA fall into several categories: re-training-based approaches [1, 2, 3, 4, 5, 6, 7], gradient-based approaches applied throughout training [8, 9, 10, 11], and gradient approaches applied only at the end [12, 13, 14, 15, 16, 17, 18, 19, 20]. We refer to the survey by [21] for a deeper look at these categories.

In this paper, rather than proposing another TDA method, we take a more reflective approach. First, we draw attention to the fact that there exist multiple *problem settings* for TDA, alongside the multiple categories of methods. These problem settings differ in the level of access assumed. In particular, we focus on what we call the "final-model-only" (FiMO) setting in which we have access only to the final trained model, and not the algorithm used to train the model or intermediate information from training (for example checkpoints). The FiMO setting is motivated by the common scenario in which TDA is performed by a different party than the one who developed the model. Models published on platforms such as HuggingFace have now made this scenario ubiquitous.

We find that since the TDA literature has not clearly differentiated these problem settings, it is also not clear on what should be the goal for TDA in the FiMO setting, and accordingly, what could be an ideal "gold standard" method. Having such a goal and gold standard (as opposed to proxy tasks such as mislabelled example detection) facilitates the development and evaluation of more practical methods. We thus reframe the problem in the FiMO setting as one of quantifying *sensitivity* of the

39th Conference on Neural Information Processing Systems (NeurIPS 2025).

Table 1: Comparison with other retrospective works on TDA plus [12] (FT = Further training)

| Category | Attribute | **Ours** | [12] | [24] | [25] | [26] | [27] | [28] |
|---|---|---|---|---|---|---|---|---|
| Setting | FiMO (explicit) | ✓ | | | | | | |
| FT refinements | Training randomness | ✓ | | | | | | |
| | Non-convergence | ✓ | | | | | ✓ | ✓ |
| Derivations | Non-convexity/stationarity | ✓ | ✓ | | | | | ✓ |
| | Generalized influence functions | ✓ | | | | | | |
| Scope | # TDA methods evaluated | 8 | 2 | 2 | 3 | 2 | 2 | 2 |

model to training instances. We call the reframed problem "FiMODA" (DA for data attribution). We then propose *further training*, starting from the given final model, as a gold standard measure of sensitivity. Notably, our proposal adjusts for the effect of further training non-converged models and accounts for the randomness of neural network training algorithms. These refinements have consequences as we show in our experiments.

Like re-training[1], further training a model multiple times can be computationally prohibitive. We thus consider approximations based on first- and second-order Taylor expansions of the further training objective. In doing so, we unify several existing gradient-based methods for TDA. These methods include gradient similarity [22] (a final-checkpoint-only case of TracIn [9]) and methods based on influence functions [12, 18, 19, 23, 20]. Given that these gradient-based methods approximate further training as we show, we submit that they are more suited to the FiMO setting, rather than as approximations to re-training where their effectiveness has been questioned [24, 25]. Our derivation of gradient-based methods also does not make assumptions of convexity or stationarity that are common in the TDA literature, and we obtain generalized influence function expressions as a result.

We investigate empirically the quality of the approximations to further training provided by different gradient-based methods. Our experiments span the modalities of tabular, image, and text data. Overall, we find that first-order gradient-based methods can give good initial approximations to further training, but the quality of approximation decays with the amount of further training. In contrast, the approximation quality of influence function methods is more persistent, but somewhat surprisingly, never as high as first-order methods at their peak. We provide code to help reproduce our experiments at `https://github.com/IBM/fimoda`.

Our contributions are summarized as follows and in Table 1: 1) We highlight the FiMO setting for TDA (Section 2). 2) We reframe the TDA problem in the FiMO setting as one of quantifying sensitivity. We articulate and refine a further training gold standard for this reframed problem (called FiMODA, Section 3). 3) We show how several gradient-based TDA methods approximate further training, theoretically (Section 4) and numerically (Section 6).

## 2 Problem Settings for Training Data Attribution

**Preliminaries** In all problem settings that we consider for TDA, we are given access to the training dataset $\mathcal{D} = \{z_i\}_{i=1}^{n}$ for the model, consisting of $n$ pairs $z_i = (x_i, y_i)$ of inputs $x_i \in \mathcal{X}$ and targets $y_i \in \mathcal{Y}$. A model $f(x; \theta)$ is a function $f : \mathcal{X} \to \mathcal{F}$, parameterized by $\theta \in \mathbb{R}^p$, that maps to an output space $\mathcal{F}$ (e.g., predicted logits for a classification model). The quality of each model output $f(x_i; \theta)$ with respect to target $y_i$ is measured by a loss function $\ell(f(x_i; \theta), y_i)$; we adopt the more compact notation $L(z_i; \theta) \triangleq \ell(f(x_i; \theta), y_i)$. We use $A$ to denote the *training algorithm* used to train $f(x; \theta)$ on dataset $\mathcal{D}$. We view $A$ as a function that takes $\mathcal{D}$ and initial model parameters $\theta^{(0)}$ (which may come from a pre-trained model) as input and outputs final model parameters $\theta^f = A(\mathcal{D}, \theta^{(0)})$.

Given a test instance $z = (x, y)$ and an *evaluation function* $g(z, \theta)$ that depends in general on both $f(x; \theta)$ and $y$, the task of TDA is to assign scores $a_i$ quantifying the importance of each training instance $z_i$ to the output $g(z, \theta)$. We refer to $a_i$ interchangeably as an attribution or influence score. Commonly, the evaluation function is the loss on $z$, $g(z, \theta) = L(z; \theta)$, or the output of the model alone, $g(z, \theta) = f(x; \theta)$. One may also consider evaluation functions that sum over test instances $z$.

**Three problem settings** We distinguish three problem settings for TDA, based on the level of access to the above-defined quantities:

---

[1]Throughout the paper, we reserve the term "re-training" to mean re-training from scratch.

1. **Training Algorithm Available (TAA)**: In this first case, we have access to the training algorithm $A$. TDA can therefore be done by re-training the model on (i.e., applying $A$ to) different subsets of $\mathcal{D}$ and evaluating the resulting effects. This scenario typically occurs when the parties training the model and performing data attribution are the same.
2. **Checkpoints Available (CPA)**: We do not have access to $A$ but do have intermediate information from the training process, for example intermediate model parameters $\boldsymbol{\theta}^{(t)}$ (i.e., "checkpoints") or gradients $\nabla_{\boldsymbol{\theta}} f(\boldsymbol{x}_i; \boldsymbol{\theta}^{(t)})$, as well as algorithm details such as learning rates. Here, TDA can be done by "tracing" the effect of training instances throughout the process. This scenario is also more typical when model training and data attribution are performed by the same party.
3. **Final Model Only (FiMO)**: We have access only to the final model $f(\boldsymbol{x}; \boldsymbol{\theta}^f)$ and not the training algorithm $A$ or intermediate information. The algorithm $A$ may actually be unavailable, or it may be available but too computationally expensive or otherwise undesirable to re-run. This scenario typically occurs when model training and data attribution are performed by different parties.

The three settings also differ in the object being attributed. In the TAA setting, it is the algorithm $A$ and its sensitivity to different training instances $\boldsymbol{z}_i$. In the CPA setting, it is the training trajectory. The FiMO setting is the most focused on the final model $f(\boldsymbol{x}; \boldsymbol{\theta}^f)$ that will actually be used.

In the remainder of the paper, we focus on the FiMO setting as it requires the least amount of access and is thus the most widely applicable. It applies for example to open-weights models that are freely downloadable from platforms such as HuggingFace. By extension, the FiMO setting applies in cases where methods for the TAA and CPA settings (i.e., re-training and tracing methods) cannot be used.[2] We will use the term "FiMODA" to refer to TDA in the FiMO setting and to distinguish it from the standard TDA problem.

## 3 A Further Training Gold Standard for Final-Model-Only Data Attribution

We first consider the question of how FiMODA should ideally be done, i.e., what should be the goal and what could serve as a "gold standard" method, where the gold standard also respects the FiMO constraint. While a gold standard method may be very expensive to carry out in many cases, it is nevertheless useful in evaluating and developing approximate FiMODA methods that are more practical. However, since the TDA literature has not explicitly delineated the FiMO setting to our knowledge, it is also not clear what the goal and gold standard should be.

Returning for a moment to the TAA setting, the natural question to ask there is one of *contribution*: how much does a training instance $\boldsymbol{z}_i$ contribute to the model through the training process? Accordingly, variants of re-training, designed to estimate these contributions, are accepted as gold standards. The simplest of these is leave-one-out (LOO) re-training where each instance $\boldsymbol{z}_i$ is left out of the training set in turn to assess its contribution. For FiMODA however, it is not clear how one can "go back in time" to determine contributions to the final model. Instead, we change the question to one of *sensitivity*: how sensitive is the given model to a training instance $\boldsymbol{z}_i$? To measure these sensitivities, we propose *further training* as a gold standard, in a manner analogous to how re-training is used in the TAA setting.

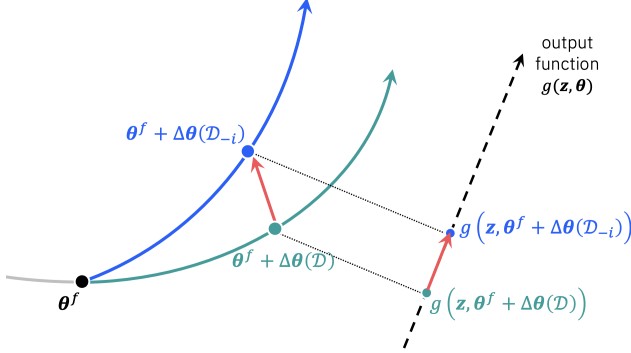

Figure 1: Given only a final model with parameters $\boldsymbol{\theta}^f$, the proposed *further training* gold standard measures the model's *sensitivity* to training sample $i$ by further training on the full training set $\mathcal{D}$ as well as leaving out sample $i$ ($\mathcal{D}_{-i}$), resulting in changed parameters $\boldsymbol{\theta}^f + \Delta\boldsymbol{\theta}(\mathcal{D})$ and $\boldsymbol{\theta}^f + \Delta\boldsymbol{\theta}(\mathcal{D}_{-i})$. The difference in outputs $g(\boldsymbol{z}, \boldsymbol{\theta}^f + \Delta\boldsymbol{\theta}(\mathcal{D}_{-i})) - g(\boldsymbol{z}, \boldsymbol{\theta}^f + \Delta\boldsymbol{\theta}(\mathcal{D}))$ indicates the sensitivity to $i$. For stochastic training, this process is repeated to obtain the expected sensitivity.

---

[2]Naturally, if the increased access of the TAA and CPA settings is available, then methods that take advantage of this can do more than is possible in the FiMO setting.

Further training can be described generically as follows. We start from the given parameters $\boldsymbol{\theta}^f$ and train on a dataset $\mathcal{D}'$ using an algorithm $A'$, i.e., $\boldsymbol{\theta}^f + \Delta\boldsymbol{\theta} = A'(\mathcal{D}', \boldsymbol{\theta}^f)$, where $\mathcal{D}'$ and $A'$ are generally different from $\mathcal{D}$ and $A$. In this work, we restrict attention to LOO further training, corresponding to LOO datasets $\mathcal{D}' = \mathcal{D}_{-i} \triangleq \mathcal{D} \setminus \{\boldsymbol{z}_i\}$ as well as $\mathcal{D}' = \mathcal{D}$. This restriction aligns with the gradient-based methods discussed in Section 4; more general perturbed datasets $\mathcal{D}'$ are of course possible. As for $A'$, our desire is for it to be representative of neural network (NN) training. We thus assume that $A'$ seeks to minimize the empirical risk $R(\mathcal{D}'; \boldsymbol{\theta}) \triangleq \sum_{\boldsymbol{z}_i \in \mathcal{D}'} L(\boldsymbol{z}_i; \boldsymbol{\theta})$:

$$\Delta\boldsymbol{\theta}(\mathcal{D}') \underset{\approx}{\in} \arg\min_{\Delta\theta} R(\mathcal{D}'; \boldsymbol{\theta}^f + \Delta\boldsymbol{\theta}), \tag{1}$$

where $\underset{\approx}{\in} \arg\min$ means "an approximate minimizer" (note that (1) may not have a unique minimizer), the notation $\boldsymbol{\theta}^f + \Delta\boldsymbol{\theta}$ is meant to indicate that the optimization is initialized at $\boldsymbol{\theta}^f$ ($\Delta\boldsymbol{\theta} = \mathbf{0}$), and $\Delta\boldsymbol{\theta}(\mathcal{D}')$ is the change in parameters resulting from applying $A'$ to $(\mathcal{D}', \boldsymbol{\theta}^f)$.

We now discuss refinements to further training to address two issues inherent to typical NN training.[3]

**Non-convergence:** The "final" parameters $\boldsymbol{\theta}^f$ are often not a stationary point of the empirical risk. Hence, as shown in Figure 1, further training on the original training set $\mathcal{D}' = \mathcal{D}$ generally yields a non-zero change $\Delta\boldsymbol{\theta}(\mathcal{D})$ and a corresponding change in the evaluation function $g(\boldsymbol{z}, \boldsymbol{\theta}^f + \Delta\boldsymbol{\theta}(\mathcal{D}))$. We refer to this as the *effect due to further training alone*. Similarly, $\Delta\boldsymbol{\theta}(\mathcal{D}_{-i})$ and $g(\boldsymbol{z}, \boldsymbol{\theta}^f + \Delta\boldsymbol{\theta}(\mathcal{D}_{-i}))$ are the effects of further training without instance $\boldsymbol{z}_i$. The difference in outputs,

$$g(\boldsymbol{z}, \boldsymbol{\theta}^f + \Delta\boldsymbol{\theta}(\mathcal{D}_{-i})) - g(\boldsymbol{z}, \boldsymbol{\theta}^f + \Delta\boldsymbol{\theta}(\mathcal{D})), \tag{2}$$

can therefore be interpreted as the sensitivity to the presence of $\boldsymbol{z}_i$, where we have adjusted for the effect of further training alone.

**Stochasticity of training:** The training algorithm $A'$ also depends on random elements, notably the order of instances in each training epoch. Denoting these random elements as $\xi$, the change in parameters becomes a function of $\xi$, $\boldsymbol{\theta}^f + \Delta\boldsymbol{\theta}(\mathcal{D}', \xi) = A'(\mathcal{D}', \boldsymbol{\theta}^f, \xi)$, as does the evaluation function $g(\boldsymbol{z}, \boldsymbol{\theta}^f + \Delta\boldsymbol{\theta}(\mathcal{D}', \xi))$. We view $\xi$ as a nuisance parameter, an artifact of the stochastic nature of modern training algorithms. Thus, we would ideally like to take an expectation over $\xi$. Applying this to the difference in (2) yields

$$a_i^* = \mathbb{E}\left[g(\boldsymbol{z}, \boldsymbol{\theta}^f + \Delta\boldsymbol{\theta}(\mathcal{D}_{-i}, \xi)) - g(\boldsymbol{z}, \boldsymbol{\theta}^f + \Delta\boldsymbol{\theta}(\mathcal{D}, \xi))\right] \tag{3}$$

as the *adjusted* and *expected* sensitivity to leaving out instance $\boldsymbol{z}_i$. Equation (3) is our proposed gold standard attribution score based on further training.

As mentioned above, further training is a natural analogue of re-training from scratch, and as shown in the next section, it is approximated by several gradient-based TDA methods. To further support the validity of further training, Appendix E.2 shows that the sensitivities quantified by further training can also detect mislabelled examples and provide insights into model behavior.

## 4 Gradient-Based Methods as Approximate Further Training

The further training gold standard described in the previous section is computationally expensive. In the case of LOO further training, the number of further trainings scales with the training set size $n$, or at least the number of training instances for which we wish to estimate attribution scores. If we wish to approximate the expectation in (3) by averaging over multiple realizations of $\xi$, that further scales the cost. It is natural therefore to consider approximate further training. In this section, we assume that the amount of further training is limited, so that the resulting changes in model parameters, $\Delta\boldsymbol{\theta}$, are small. We show that several gradient-based methods for TDA can be re-derived from this perspective. We argue therefore that they are better viewed as approximations to further training and better suited to FiMODA than to other TDA variants.

It is important to note that unlike the standard influence function literature, we do not assume that the empirical risk $R(\mathcal{D}', \boldsymbol{\theta})$ is convex, nor do we assume that the given parameters $\boldsymbol{\theta}^f$ are a stationary point of $R(\mathcal{D}, \boldsymbol{\theta})$ over the full training set $\mathcal{D}$. As a result of the latter, we obtain generalized influence function expressions (Proposition 1 and Corollary 2). While we then specialize to the standard expressions used in existing works, the generalized expressions could be useful in future work.

---

[3]For convex training objectives, these issues are not much of a concern (please see Appendix A).

## 4.1 Approximate further training

The assumption of small $\Delta\boldsymbol{\theta}$ leads us to consider first- and second-order Taylor expansions of (1):

$$\widehat{\Delta\boldsymbol{\theta}}(\mathcal{D}') = \underset{\Delta\boldsymbol{\theta}}{\arg\min}\, R(\mathcal{D}';\boldsymbol{\theta}^f) + \left(\nabla_{\boldsymbol{\theta}} R(\mathcal{D}';\boldsymbol{\theta}^f)\right)^T \Delta\boldsymbol{\theta} + \frac{1}{2}\Delta\boldsymbol{\theta}^T \nabla_{\boldsymbol{\theta}}^2 R(\mathcal{D}';\boldsymbol{\theta}^f)\Delta\boldsymbol{\theta} + \frac{\lambda}{2}\|\Delta\boldsymbol{\theta}\|_2^2, \quad (4)$$

where $\nabla_{\boldsymbol{\theta}} R(\mathcal{D}';\boldsymbol{\theta}^f)$ and $\nabla_{\boldsymbol{\theta}}^2 R(\mathcal{D}';\boldsymbol{\theta}^f)$ denote the gradient and Hessian of the empirical risk with respect to $\boldsymbol{\theta}$ evaluated at $\boldsymbol{\theta} = \boldsymbol{\theta}^f$, and the first-order expansion omits the Hessian term. In both cases, an $\ell_2$ regularizer $(\lambda/2)\|\Delta\boldsymbol{\theta}\|_2^2$ has been added in (4), corresponding to the "damping" term in influence function estimation. As discussed in Appendix B.1, the regularizer can be motivated in two ways: 1) enforcing the smallness of $\Delta\boldsymbol{\theta}$ and the accuracy of the Taylor expansion; 2) ensuring that solutions to (4) do not diverge. To achieve 2), we make the following assumption.

**Assumption 1.** If the Hessian $\nabla_{\boldsymbol{\theta}}^2 R(\mathcal{D}';\boldsymbol{\theta}^f)$ is present in (4), then $\lambda > -\lambda_{\min}(\nabla_{\boldsymbol{\theta}}^2 R(\mathcal{D}';\boldsymbol{\theta}^f))$, where $\lambda_{\min}$ denotes the minimum eigenvalue.

We may similarly expand the evaluation function $g(\boldsymbol{z},\boldsymbol{\theta}^f + \Delta\boldsymbol{\theta})$ to first order in $\Delta\boldsymbol{\theta}$. Doing so reduces the difference in (2) to the inner product

$$\hat{a}_i = \left(\nabla_{\boldsymbol{\theta}} g(\boldsymbol{z},\boldsymbol{\theta}^f)\right)^T \left(\widehat{\Delta\boldsymbol{\theta}}(\mathcal{D}_{-i}) - \widehat{\Delta\boldsymbol{\theta}}(\mathcal{D})\right). \quad (5)$$

## 4.2 First-order methods

In the case of first-order Taylor expansion, the absence of the Hessian term in (4) makes its solution straightforward. For $\mathcal{D}' = \mathcal{D}$, we have $\widehat{\Delta\boldsymbol{\theta}}(\mathcal{D}) = -\lambda^{-1}\nabla_{\boldsymbol{\theta}} R(\mathcal{D},\boldsymbol{\theta}^f)$, and similarly $\widehat{\Delta\boldsymbol{\theta}}(\mathcal{D}_{-i}) = -\lambda^{-1}(\nabla_{\boldsymbol{\theta}} R(\mathcal{D},\boldsymbol{\theta}^f) - \nabla_{\boldsymbol{\theta}} L(\boldsymbol{z}_i,\boldsymbol{\theta}^f))$. Hence (5) becomes

$$\hat{a}_i = \lambda^{-1}\left(\nabla_{\boldsymbol{\theta}} g(\boldsymbol{z},\boldsymbol{\theta}^f)\right)^T \nabla_{\boldsymbol{\theta}} L(\boldsymbol{z}_i,\boldsymbol{\theta}^f), \quad (6)$$

proportional to the inner product between training loss and test evaluation gradients. This corresponds to the "**Grad-Dot**" method [22], and to a special case of TracIn [9] that uses only the final checkpoint. A variation is to use cosine similarity ("**Grad-Cos**") instead of the unnormalized inner product.

## 4.3 Influence function methods

To show that the approximate further training formulation in (4), (5) recovers influence function-based TDA methods, we first present two ingredients: influence functions for (4), and the Gauss-Newton approximation to the Hessian that is commonly made.

### 4.3.1 Influence functions for (4)

Influence functions approximate the effect of down-weighting the training loss of instance $i$ by a small amount $\epsilon$, i.e., of changing the empirical risk from the full-dataset risk $R(\mathcal{D};\boldsymbol{\theta})$ to $R(\mathcal{D};\boldsymbol{\theta}) - \epsilon L(\boldsymbol{z}_i;\boldsymbol{\theta})$. We use $\mathcal{D}_{-i,\epsilon}$ to denote this down-weighted dataset. We apply the implicit function theorem to the stationary point condition for (4) to derive the following (proof in Appendix B.2).

**Proposition 1.** Given Assumption 1, the parameter change due to down-weighting training instance $\boldsymbol{z}_i$ by an amount $\epsilon$ is approximated by influence functions as

$$\widehat{\Delta\boldsymbol{\theta}}(\mathcal{D}_{-i,\epsilon}) - \widehat{\Delta\boldsymbol{\theta}}(\mathcal{D}) \approx \epsilon\left(\boldsymbol{H}(\boldsymbol{\theta}^f) + \lambda\boldsymbol{I}\right)^{-1}\left(\nabla_{\boldsymbol{\theta}} L(\boldsymbol{z}_i;\boldsymbol{\theta}^f) + \nabla_{\boldsymbol{\theta}}^2 L(\boldsymbol{z}_i;\boldsymbol{\theta}^f)\widehat{\Delta\boldsymbol{\theta}}(\mathcal{D})\right), \quad (7)$$

where $\boldsymbol{H}(\boldsymbol{\theta}^f) = \nabla_{\boldsymbol{\theta}}^2 R(\mathcal{D};\boldsymbol{\theta}^f)$ is the Hessian of the full-dataset empirical risk evaluated at $\boldsymbol{\theta}^f$.

If $\boldsymbol{\theta}^f$ is a stationary point of the full-dataset risk $R(\mathcal{D};\boldsymbol{\theta})$, then $\Delta\boldsymbol{\theta}(\mathcal{D}) = \widehat{\Delta\boldsymbol{\theta}}(\mathcal{D}) = \boldsymbol{0}$ and (7) reduces to the familiar inverse Hessian-gradient product (with damping):

$$\widehat{\Delta\boldsymbol{\theta}}(\mathcal{D}_{-i,\epsilon}) \approx \epsilon\left(\boldsymbol{H}(\boldsymbol{\theta}^f) + \lambda\boldsymbol{I}\right)^{-1}\nabla_{\boldsymbol{\theta}} L(\boldsymbol{z}_i;\boldsymbol{\theta}^f). \quad (8)$$

If $\boldsymbol{\theta}^f$ is not stationary however, then $\widehat{\Delta\boldsymbol{\theta}}(\mathcal{D}) \neq \boldsymbol{0}$ as discussed in Section 3. Previous works [12] neglect the last $\widehat{\Delta\boldsymbol{\theta}}(\mathcal{D})$ term in (7), arguing that if $\boldsymbol{\theta}^f$ is near-stationary, then $\widehat{\Delta\boldsymbol{\theta}}(\mathcal{D})$ is small and the product $\epsilon\widehat{\Delta\boldsymbol{\theta}}(\mathcal{D})$ is second-order. It is also convenient to neglect this term to avoid computing $\widehat{\Delta\boldsymbol{\theta}}(\mathcal{D})$. We will neglect it as well in the main paper to be consistent with previous TDA methods. In Appendices B.2 and B.3 however, we provide an expression for $\widehat{\Delta\boldsymbol{\theta}}(\mathcal{D})$ and also show how to account for the $\widehat{\Delta\boldsymbol{\theta}}(\mathcal{D})$ term in (7) without requiring the Hessian $\nabla_{\boldsymbol{\theta}}^2 L(\boldsymbol{z}_i;\boldsymbol{\theta}^f)$.

### 4.3.2 Gauss-Newton approximate Hessian

It is often the case that the loss function $L(z_i; \theta)$ is a composition $\bar{\ell}(\bar{f}(z_i; \theta))$ of a *scalar-valued* model output function $\bar{f}(z_i; \theta)$ with a convex univariate loss function $\bar{\ell}(\bar{f})$. Here we allow $\bar{f}(z_i; \theta)$ to also depend on the target $y_i$. For example, if $f(x_i; \theta)$ is a regression model, $\bar{f}(z_i; \theta) = f(x_i; \theta) - y_i$ can be the prediction error. For multi-class classification, [19] define $\bar{f}(z_i; \theta)$ to be the logit of the predicted probability of the target class $y_i$, which allows multi-class classification to be handled in the same way (see [19, Sec. 3.3] for details).

Given the composition $L(z_i; \theta) = \bar{\ell}(\bar{f}(z_i; \theta))$, the gradient and Hessian of $L(z_i; \theta)$ are given by

$$\nabla_\theta L(z_i; \theta) = \bar{\ell}'(\bar{f}(z_i; \theta)) \nabla_\theta \bar{f}(z_i; \theta), \tag{9}$$

$$\nabla_\theta^2 L(z_i; \theta) = \bar{\ell}''(\bar{f}(z_i; \theta)) \nabla_\theta \bar{f}(z_i; \theta) \nabla_\theta \bar{f}(z_i; \theta)^T + \bar{\ell}'(\bar{f}(z_i; \theta)) \nabla_\theta^2 \bar{f}(z_i; \theta). \tag{10}$$

The Gauss-Newton approximation to the Hessian drops the second term in (10), which requires the Hessian $\nabla_\theta^2 \bar{f}(z_i; \theta)$ that is more difficult to compute.

To obtain more compact expressions, let us define the gradient vectors $g_i = \nabla_\theta \bar{f}(z_i; \theta^f)$, $i = 1, \ldots, n$ and matrix $G = [g_1 \quad \ldots \quad g_n]^T$, vector $r = -\left[ \bar{\ell}'(\bar{f}(z_1; \theta^f)) \quad \ldots \quad \bar{\ell}'(\bar{f}(z_n; \theta^f)) \right]^T$, and $n \times n$ diagonal matrix $V$ with $\bar{\ell}''(\bar{f}(z_1; \theta^f)), \ldots, \bar{\ell}''(\bar{f}(z_n; \theta^f))$ as its diagonal entries. Using these definitions, (9), and the Gauss-Newton simplification of (10), and disregarding the constant term, the quadratic optimization in (4) can be rewritten for $\mathcal{D}' = \mathcal{D}$ as

$$\min_{\Delta\theta} \; -r^T G \Delta\theta + \frac{1}{2}\Delta\theta^T (G^T V G + \lambda I)\Delta\theta. \tag{11}$$

The same derivation of influence functions in the proof of Proposition 1 applies to the Gauss-Newton approximation. Effectively, (9) and (10) (with the Gauss-Newton simplification) are used to substitute for the gradient and Hessian in (7).

**Corollary 2.** Under the Gauss-Newton approximation, the influence function in Proposition 1 becomes

$$\widehat{\Delta\theta}(\mathcal{D}_{-i,\epsilon}) - \widehat{\Delta\theta}(\mathcal{D}) \approx \epsilon \left( v_{ii} g_i^T \widehat{\Delta\theta}(\mathcal{D}) - r_i \right) (G^T V G + \lambda I)^{-1} g_i.$$

As with Proposition 1, this is a generalization of the standard influence function to non-stationary $\theta^f$ and non-zero $\widehat{\Delta\theta}(\mathcal{D})$. It reduces to the standard form used in previous works when $\widehat{\Delta\theta}(\mathcal{D}) = 0$:

$$\widehat{\Delta\theta}(\mathcal{D}_{-i,\epsilon}) \approx -\epsilon r_i (G^T V G + \lambda I)^{-1} g_i. \tag{12}$$

### 4.3.3 Relationships with existing influence function methods

We now discuss how existing influence function methods correspond to the simplified versions (8) and (12) of Proposition 1 and Corollary 2. We refer to the cited works for more details on the methods.

**Conjugate gradient (CG) and LiSSA** Both (8) and (12) require computing an inverse Hessian-gradient product, with either the damped true Hessian $H(\theta^f) + \lambda I$ or the Gauss-Newton Hessian $G^T V G + \lambda I$. For models with a large number of parameters $p$, the $O(p^3)$ computational cost of directly solving the system of equations is prohibitive. [12] proposed to apply CG [29] and LiSSA [30], two methods that approximate the solution iteratively. The former does so by applying the CG algorithm to an equivalent quadratic minimization problem, while the latter uses a truncated Neumann series. Both can be applied to either (8) or (12).

The following methods are based on and therefore specific to the Gauss-Newton approximation.

**TRAK** TRAK [19] can be seen as a hybrid TDA method that combines a gradient-based approximation with re-training (training an ensemble of models). For FiMODA where re-training is not possible, we focus on the gradient-based part of TRAK by setting its parameter $M$ (number of trained models) to 1. This variant, which we refer to as $\text{TRAK}_{M=1}$, falls under the Gauss-Newton framework of Section 4.3.2. First, TRAK applies a random projection matrix $P \sim \mathcal{N}(0, 1)^{p \times k}$ to reduce the dimensionality of the gradients: $\phi_i = P^T g_i$ and $\Phi = GP$. With $\phi_i$ and $\Phi$ in place of $g_i$ and $G$, $\text{TRAK}_{M=1}$ corresponds to a special case of (12) with $\lambda = 0$ (no regularization) and $V = I$ (non-identity $V$ found empirically to have little effect), i.e., $\widehat{\Delta\theta}_{\text{TRAK}}(\mathcal{D}_{-i}) = -r_i (\Phi^T \Phi)^{-1} \phi_i$.

**EK-FAC**   [23] start with the Gauss-Newton approximation in their work, corresponding to (12). Next, they adopt the K-FAC and EK-FAC approximations from [31, 32], which first approximate the Gauss-Newton Hessian as block-diagonal, where each block corresponds to a layer in a NN. This allows influence functions to be computed separately for each layer. Then for certain layers, an uncorrelatedness assumption is made that permits a Kronecker factorization of the corresponding block of the Hessian, making inversion much more efficient.

**DataInf**   [20] start with a variant of the Gauss-Newton approximation in which $\bar{\ell}(\bar{f}) = \bar{f}$ is the identity function and $\boldsymbol{g}_i = \nabla_{\boldsymbol{\theta}} L(\boldsymbol{z}_i; \boldsymbol{\theta}^f)$ is a gradient of the loss. They then make the same block-diagonal/layer-wise approximation as [23]. Setting this simplification aside, DataInf corresponds to (12) with $r_i = -1$ (identity $\bar{\ell}$) and $\boldsymbol{V} = (1/n)\boldsymbol{I}$ (identity $\ell$, average over $\mathcal{D}$ instead of sum). The result is $\widehat{\Delta \boldsymbol{\theta}(\mathcal{D}_{-i})} = ((1/n)\boldsymbol{G}^T\boldsymbol{G} + \lambda\boldsymbol{I})^{-1}\boldsymbol{g}_i$. The key idea of DataInf is to then interchange the order of averaging and matrix inversion (please see [20]).

# 5   Related Work

This work builds upon existing works that also take a retrospective look at TDA methods, and influence functions especially [24, 26, 27, 28, 25]. We discuss individual papers more in Appendix C. Here, we note key differences between our work and these prior works (summarized in Table 1):

1. **More gradient-based methods**: Our unified view (Section 4) and numerical comparisons (Section 6) encompass gradient-based methods beyond influence functions, including more recent ones such as TRAK and DataInf as well as first-order methods. We thereby obtain insights into similarities and differences among gradient-based methods, in addition to how well they approximate further training. In contrast, [24, 26, 27, 28] mainly focus on influence functions, while [25] evaluate fewer methods than we do.
2. **Further training**: The further training gold standard in Section 3 formalizes and refines similar procedures used in previous works (in Appendix D.2 in [27], in Section 5.2 of [28]). The proposal of averaging over realizations of stochastic further training appears to be new, and our results in Figures 3 and 7 show that it brings further training closer to what gradient-based methods estimate. [27] propose an alternative gold standard called the proximal Bregman response function (PBRF). The PBRF however involves a non-standard Bregman distance objective, tailored specifically to be closer to influence functions, whereas our further training is more generic.
3. **FiMO setting**: We explicitly define the FiMO setting, which these previous works have not.

# 6   Numerical Comparison of Gradient-Based Methods to Further Training

In Section 4, we showed that many gradient-based TDA methods are approximations to further training for FiMODA. We now report on experiments that assess the quality of these approximations. Code to help reproduce these experiments is provided at `https://github.com/IBM/fimoda`.

## 6.1   Experimental setup

We refer to Appendix D for more details on data, training, computation, the evaluation metric, etc.

**Datasets**   Our experiments span the modalities of tabular, image, and text data. However, since we aimed to implement the further training gold standard described in Section 3, which is computationally expensive, our experiments are most comprehensive in the tabular case. We used four tabular datasets: two for regression, Concrete Strength and Energy Efficiency from the UCI repository [33] following [27], and two larger ones for classification, FICO Challenge [34] and Folktables [35]. For image data, we chose the CIFAR-10 image classification dataset [36], while for text, we used the SST-2 sentiment classification dataset [37], which is part of the GLUE benchmark [38].

**"Final" Models**   For all tabular datasets, we used a 2-hidden-layer multi-layer perceptron (MLP) with 128 units in each hidden layer. The MLPs were trained using stochastic gradient descent (SGD) to yield the final model $f(\boldsymbol{x}, \boldsymbol{\theta}^f)$. For CIFAR-10, we used a ResNet-9 architecture [39] and trained it using SGD. For SST-2, we fine-tuned a pre-trained BERT model [40] using AdamW [41].

**Selection of test and training instances**   Our aim was to estimate influence scores of a subset $\mathcal{L} \subset \mathcal{D}$ of training instances on the losses of a subset of $m$ test instances. Thus, the evaluation

function was the loss, $g(\boldsymbol{z}, \boldsymbol{\theta}) = L(\boldsymbol{z}; \boldsymbol{\theta})$. For the Concrete and Energy datasets, we considered all test instances, while for the other datasets, we selected $m = 100$ test instances at random. The training subset $\mathcal{L}$ was also selected randomly with $l = |\mathcal{L}| = 100$ for the tabular datasets and $l = 50$ for CIFAR-10 and SST-2. The training subset size $l$ was limited by our aim to conduct further training, since the number of further trainings is proportional to $l$. Nevertheless, $l = 50$ or $100$ is greater than the number of training instances used in similar retrospective work on influence functions ([27] used 20 instances according to their Appendix C.2, [28] used 32 in their Section 5.2).

**Implementation of further training**     For the results in this section, the further training algorithm $A'$ is the same as $A$, i.e., AdamW for BERT and SGD otherwise. Appendix E.1 has results on the tabular datasets using Adam as $A'$ instead. We evaluated the approximation quality of gradient-based methods as a function of the amount of further training (epochs or steps). To approximate the expectation over $\xi$ in (3), we took averages over $r = 100$ random seeds (denoted as realizations $\xi^{(s)}$ in (13) below), which control the order of instances in each training epoch. To achieve the effect of leaving out a training instance $\boldsymbol{z}_i$, we followed [28] in subtracting a second loss term $L(\boldsymbol{z}_i; \boldsymbol{\theta})/n$ from the usual loss for *every* batch (please see Appendix D.3 for explanation of the $1/n$ scaling). While [28] did not provide a justification for this practice, we believe that it also mitigates the randomness of stochastic mini-batch training, since the mini-batch that $\boldsymbol{z}_i$ appears in/is left out of is random.

We considered two methods for adjusting for the effect of further training alone. The first is as specified in (2), (3), i.e., subtracting the output due to further training on the full dataset $\mathcal{D}$. We found however that for some datasets, this adjustment left a non-negligible bias that varies from epoch to epoch, resulting in noisy cosine similarities. Please see Appendices D.3 and E.1 for further discussion of and results from this first method. The second method (used in this section) subtracts the mean of the effects due to each LOO dataset $\mathcal{D}_{-i}$. Incorporating also the averaging over random seeds, the "gold" attribution score from further training is thus

$$a_i = \frac{1}{r} \sum_{s=1}^{r} \left[ g\big(\boldsymbol{z}, \boldsymbol{\theta}^f + \Delta\boldsymbol{\theta}(\mathcal{D}_{-i}, \xi^{(s)})\big) - \frac{1}{l} \sum_{i' \in \mathcal{L}} g\big(\boldsymbol{z}, \boldsymbol{\theta}^f + \Delta\boldsymbol{\theta}(\mathcal{D}_{-i'}, \xi^{(s)})\big) \right]. \tag{13}$$

**Approximate FiMODA methods**     We experimented with the first-order methods Grad-Dot [22] (final-checkpoint-only TracIn [9]) and Grad-Cos, and influence function methods CG and LiSSA [30, 12], TRAK$_{M=1}$ [19], EK-FAC [23], and DataInf [20]. For LiSSA, we considered both the Gauss-Newton version ((12), referred to simply as "LiSSA") as well as the true Hessian version ((8), LiSSA-H). For CG, only the Gauss-Newton version was used. CG could not be run on CIFAR-10 and SST-2 because it took too long, whereas LiSSA-H ran out of GPU memory on SST-2. For both CG and LiSSA, we used a damping value of $\lambda = 0.01$, which we found to yield a better approximation to further training than the $\lambda = 0.001$ value used by [27] (see Appendix D.4 for other parameter settings). We adapted TRAK for regression (see Appendix D.4).

**Evaluation metric**     For each test instance evaluated, we have a vector $\boldsymbol{a} \in \mathbb{R}^l$ of gold attribution scores from further training (13), and corresponding vectors $\hat{\boldsymbol{a}}$ from gradient-based methods. We use the cosine similarity between $\boldsymbol{a}$ and $\hat{\boldsymbol{a}}$ as the evaluation metric. The choice of cosine similarity is motivated by the following reasons elaborated on in Appendix D.5: 1) It measures similarity in signed magnitudes between $\boldsymbol{a}$ and $\hat{\boldsymbol{a}}$, which is in line with the derivations in Section 4; 2) it is not sensitive however to an overall scale difference between $\boldsymbol{a}$ and $\hat{\boldsymbol{a}}$; 3) it is a more demanding measure than Pearson correlation. Appendix E.1 presents qualitatively similar results using Spearman rank correlation, a common metric in the TDA literature.

## 6.2   Results

**Similarity vs. amount of further training**     Figure 2 shows the cosine similarity between the attribution scores of gradient-based TDA methods and further training, as a function of the amount of further training. The curves represent the mean cosine similarity over the $m$ test instances, while the shaded area corresponds to $\pm 1$ standard error. We make the following observations:

- **First-order methods:** The two first-order methods, Grad-Dot and Grad-Cos, have cosine similarity curves that decay with the amount of further training. Grad-Dot is generally superior to Grad-Cos, and the initial cosine similarity of the former is among the highest of any approximate method. This aligns with Section 4.2 showing that Grad-Dot is the direct result of first-order approximation of further training, whereas the normalization of Grad-Cos is not supported by the theory. The rate

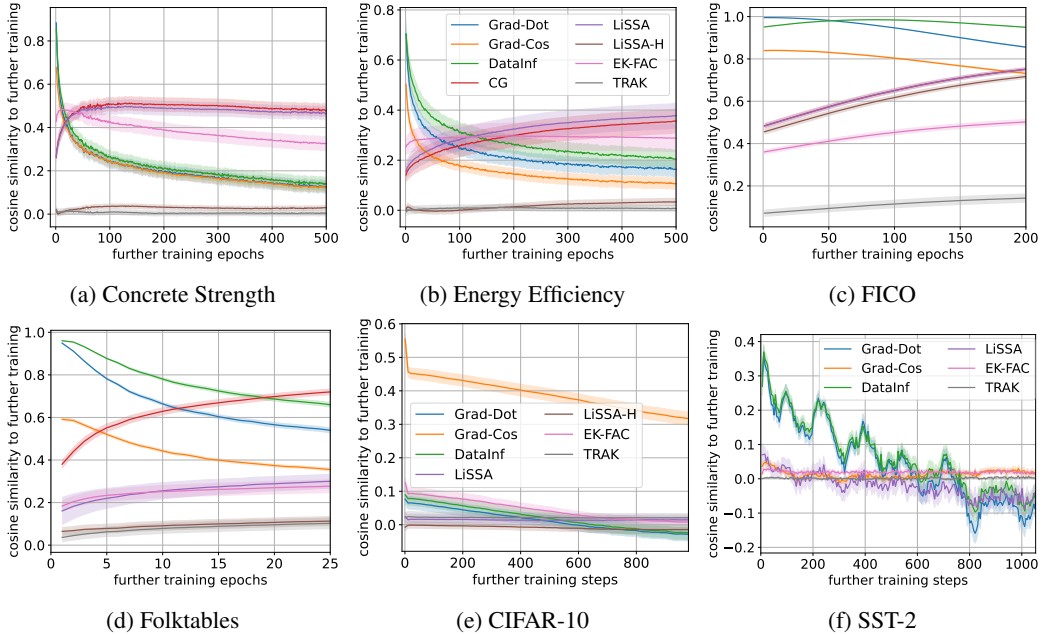

(a) Concrete Strength     (b) Energy Efficiency     (c) FICO

(d) Folktables     (e) CIFAR-10     (f) SST-2

Figure 2: Cosine similarity between attribution scores of gradient-based TDA methods and further training, as a function of the number of epochs or steps of further training. The legend in panel (b) applies to (a)–(d). In panel (a), Grad-Cos occludes Grad-Dot; in panel (c), LiSSA occludes CG.

of decay after the initial peak varies with the dataset. One reason may be how far from stationary are the final model parameters $\boldsymbol{\theta}^f$. If they are farther, then the parameter changes $\Delta\boldsymbol{\theta}(\mathcal{D})$ and $\Delta\boldsymbol{\theta}(\mathcal{D}_{-i})$ are larger, the first-order approximation is poorer, and the cosine similarity is thus lower.

- **DataInf:** We find it interesting that DataInf behaves like a first-order method, despite its attempt to incorporate second-order information. It is especially similar to Grad-Dot (cosine similarity between the two is typically above $0.95$) and generally slightly higher in similarity to further training. We further interpret this similarity between DataInf and Grad-Dot in Appendix E.1.
- **CG, LiSSA, LiSSA-H:** Unlike the first-order methods, these influence function methods have cosine similarity curves that do not decay and may even increase with further training. However, they do not attain cosine similarities as high as the first-order methods (at least within the amount of further training that we conducted). A possible interpretation is that influence function methods provide better longer-range approximations to further training by using second-order information. However, the approximation may not be especially good at any point. Among the three methods, CG and LiSSA are similar to each other in the first row of Figure 2, but LiSSA seems to degrade in the second row as the model size increases. LiSSA-H on the other hand is consistently worse.
- **EK-FAC, TRAK$_{M=1}$:** These influence function methods make additional approximations. Compared to LiSSA, EK-FAC has similar or higher cosine similarity in Figures 2d–2f, but is worse in 2a–2c. TRAK$_{M=1}$ did not give good approximations to further training (please see Appendix E.1).
- **CIFAR-10, SST-2:** The cosine similarities are significantly lower for these non-tabular datasets. The immediate reason for this is that the gold attribution vectors $\boldsymbol{a}$ behave differently than they do for tabular data, as discussed in Appendix E.1.

**Similarity vs. amount of averaging** In Figure 3, we plot the maximum cosine similarity over epochs/steps (i.e., the maximum of each curve in Figure 2) as a function of the number $r$ of random seeds averaged in (13) to obtain gold attribution values. The exact averaging procedure is given in Appendix E.1, along with a full version of Figure 3 with all datasets (Figure 7). We find in all cases that the cosine similarities increase with the number of seeds, implying that the averaging of further training runs brings it closer to what gradient-based methods estimate. This supports our proposal of averaging and its theoretical counterpart, the expectation in (3). The increases are more noticeable for the first-order methods and DataInf, and for certain datasets like SST-2.

# 7 Discussion and Future Work

**Summary** We have highlighted the final-model-only setting for TDA, recasting the problem (which we call FiMODA) as one of measuring the model's sensitivity to training instances rather than their contribution. We accordingly proposed further training, with refinements, as a gold standard method for FiMODA. We showed how several existing gradient-based TDA methods approximate further training, theoretically and numerically. These relationships suggest that gradient-based methods are better suited to FiMODA than to other TDA settings.

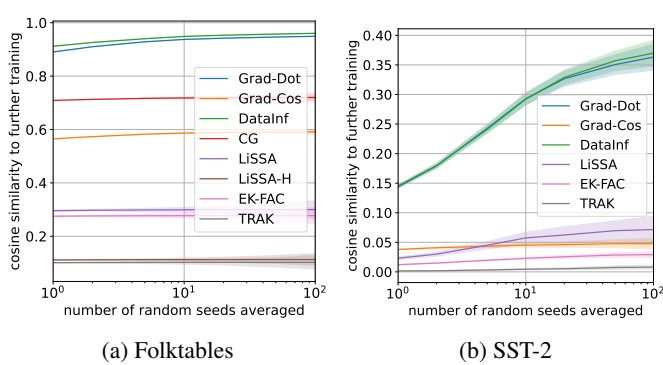

(a) Folktables      (b) SST-2

Figure 3: Maximum cosine similarity between attribution scores of gradient-based TDA methods and further training, as a function of the number of random seeds averaged.

**Limitations** The further training gold standard as described in this work is computationally expensive (as acknowledged at the beginning of Section 4). Performing further training also limited the scale of the experiments in Section 6 in terms of the models considered, number of training instances left out, etc. The computational expense might be mitigated in two ways: First, Figures 3 and 7–9 suggest that while averaging does bring further training closer to gradient-based methods, most of the gain is achieved with 10 or 20 seeds compared to 100. So reducing the number of random seeds is a computational trade-off that could be made. Second, while we have implemented further training as full fine-tuning, it is possible to use LoRA fine-tuning or other parameter-efficient methods (especially for LLMs) as another computational trade-off. The LoRA Ensemble method of [42] shows concretely how both ideas, ensembling and LoRA fine-tuning, can be brought together in the context of TDA. We mention other limitations in Appendix F.

**Discussion of experimental results** Perhaps the most salient aspect of our experimental results (Figures 2, 3 in Section 6.2 and variations in Appendix E.1) is the difference between the first-order-like methods (Grad-Dot, Grad-Cos, DataInf) and influence-function-based methods. In particular, the former can achieve significantly higher cosine similarities than the latter when the amount of further training is low. This contrasts with previous work in TDA in the TAA or CPA settings, where influence-function-based methods are generally seen as stronger [19, 11]. Our results for TRAK may be especially surprising in this regard, for which we point to two factors: First, in the FiMO setting, we have only $M = 1$ checkpoint and can only evaluate the TRAK$_{M=1}$ variant, whereas [19] shows that using an ensemble of $M \gg 1$ checkpoints greatly improves performance. Second, we compute similarity with further training attribution values, whereas [19] evaluate using their linear datamodelling score (LDS) metric, which is based on re-training and for the TAA setting.

Overall however, we think that our experimental results leave something to be desired for researchers in TDA. It could be argued that the gradient-based methods are "actually similar" to further training only for the first-order-like methods, under limited further training, and perhaps not at all in Figures 2e, 2f. (It is perhaps worth recalling that all of these gradient-based methods are existing and we are not advocating for any particular method.) Our results therefore motivate development of higher-quality approximate FiMODA methods, especially for non-tabular models. One possible starting point is the generalized influence function expressions in Proposition 1 and Corollary 2, which suggest a potential role for the parameter change $\widehat{\Delta\theta}(\mathcal{D})$ from further training on the full dataset. Moreover, Figure 2 suggests the question of whether the strengths of first- and second-order methods could be combined, i.e., higher initial similarity with less decay. We note that the damping parameter $\lambda$ could be used to interpolate between first- and second-order methods.

**Additional future work** Appendix F elaborates on the following points: 1) What is the right amount of further training to determine sensitivity to training instances? 2) Section 3 restricted attention to LOO further training to align with gradient-based methods; group influence [14, 16, 43] is worthy of further study. 3) Further training could also be applied to *new, unseen* data.

## Acknowledgements

We thank Swagatam Haldar for early discussions on the TDA literature. We also thank the anonymous NeurIPS reviewers for their constructive comments and discussions, in particular for clarifying that we are indeed reframing the problem for the FiMO setting and for suggesting an expanded discussion section.

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

## A    Further Training with Convex Objectives

For more classical models such as linear regression, logistic regression, and support vector machines, the empirical risk $R(\mathcal{D}'; \boldsymbol{\theta})$ in the further training objective (1) is a convex function of the parameters $\boldsymbol{\theta}$. If we further assume that $R(\mathcal{D}'; \boldsymbol{\theta})$ is strictly convex in $\boldsymbol{\theta}$ (which is usually the case), then the exact minimizer is unique,

$$\boldsymbol{\theta}^*(\mathcal{D}') = \arg\min_{\boldsymbol{\theta}} R(\mathcal{D}'; \boldsymbol{\theta}),$$

and barring numerical issues (for example with convex optimization solvers), we can tractably compute it. Moreover, it does not matter whether the optimization is initialized at initial parameter values $\boldsymbol{\theta}^{(0)}$ or trained parameters $\boldsymbol{\theta}^f$. Thus, further training and re-training become equivalent, provided that they are optimizing the same empirical risk.

Given the relative ease of obtaining the exact minimizer $\boldsymbol{\theta}^*(\mathcal{D})$ on the full training set, it is reasonable to assume that any final, trained parameters have converged to it, i.e., $\boldsymbol{\theta}^f = \boldsymbol{\theta}^*(\mathcal{D})$. In this case, there is no non-convergence problem as in Section 3, and further training would yield no change, i.e., $\Delta\boldsymbol{\theta}(\mathcal{D}) = \boldsymbol{0}$ in (1). As for stochastic training algorithms, they can of course also be applied to convex objectives, often with convergence guarantees (see e.g. [44] as one example among many). This greatly reduces the problem of stochasticity.

## B    Approximate Further Training

### B.1    Regularization/damping term

The following reasons provide optimization-based justification for the $\ell_2$ regularizer $(\lambda/2)\|\Delta\boldsymbol{\theta}\|_2^2$ added in (4). This corresponds to the "damping" term $\lambda\boldsymbol{I}$ added to the Hessian in estimating influence functions, where it is usually motivated simply as a means to ensure invertibility/positive definiteness.

1. **Enforce assumption/trust region:** Penalizing the $\ell_2$ norm of $\Delta\boldsymbol{\theta}$ enforces the assumption that $\Delta\boldsymbol{\theta}$ is small. It can also be viewed as enforcing a *trust region* [45], a region around $\boldsymbol{\theta}^f$ where the second-order expansion gives a better approximation.

2. **Ensure optimization is well-posed:** The loss function $L(\boldsymbol{z}; \boldsymbol{\theta})$ for neural networks is typically non-convex, which implies that the empirical risk Hessian $\nabla_{\boldsymbol{\theta}}^2 R(\mathcal{D}'; \boldsymbol{\theta}^f)$ may not be positive semidefinite. In the absence of the regularizer, if $\nabla_{\boldsymbol{\theta}}^2 R(\mathcal{D}'; \boldsymbol{\theta}^f)$ has a negative eigenvalue, then the objective value in (4) would tend to $-\infty$ in the direction of the corresponding eigenvector. If we add regularization with $\lambda$ larger in magnitude than the most negative eigenvalue of $\nabla_{\boldsymbol{\theta}}^2 R(\mathcal{D}'; \boldsymbol{\theta}^f)$, then such divergence is prevented and an optimal solution to (4) exists. This consideration leads to Assumption 1.

### B.2    Proofs

*Proof of Proposition 1.* We substitute the down-weighted risk $R(\mathcal{D}; \boldsymbol{\theta}) - \epsilon L(\boldsymbol{z}_i; \boldsymbol{\theta})$ for $R(\mathcal{D}; \boldsymbol{\theta})$ everywhere in (4). Then the stationary point (i.e., zero-gradient) condition for (4) becomes

$$F(\epsilon, \Delta\boldsymbol{\theta}) \triangleq \nabla_{\boldsymbol{\theta}} R(\mathcal{D}; \boldsymbol{\theta}^f) - \epsilon \nabla_{\boldsymbol{\theta}} L(\boldsymbol{z}_i; \boldsymbol{\theta}^f) + \left(\boldsymbol{H}(\boldsymbol{\theta}^f) - \epsilon \nabla_{\boldsymbol{\theta}}^2 L(\boldsymbol{z}_i; \boldsymbol{\theta}^f) + \lambda I\right) \Delta\boldsymbol{\theta} = \boldsymbol{0}. \quad (14)$$

For $\epsilon = 0$, i.e., no down-weighting, (14) is satisfied by the full-dataset parameter change

$$\widehat{\Delta\boldsymbol{\theta}}(\mathcal{D}) = -\left(\boldsymbol{H}(\boldsymbol{\theta}^f) + \lambda\boldsymbol{I}\right)^{-1} \nabla_{\boldsymbol{\theta}} R(\mathcal{D}; \boldsymbol{\theta}^f). \quad (15)$$

For small $\epsilon$, we apply the implicit function theorem to obtain an expression for $\widehat{\Delta\boldsymbol{\theta}}(\mathcal{D}_{-i,\epsilon})$. The relevant Jacobians of function $F$ defined in (14) are

$$\boldsymbol{J}_{F,\Delta\boldsymbol{\theta}} = \boldsymbol{H}(\boldsymbol{\theta}^f) - \epsilon \nabla_{\boldsymbol{\theta}}^2 L(\boldsymbol{z}_i; \boldsymbol{\theta}^f) + \lambda\boldsymbol{I},$$

$$\boldsymbol{J}_{F,\epsilon} = -\nabla_{\boldsymbol{\theta}} L(\boldsymbol{z}_i; \boldsymbol{\theta}^f) - \nabla_{\boldsymbol{\theta}}^2 L(\boldsymbol{z}_i; \boldsymbol{\theta}^f) \Delta\boldsymbol{\theta},$$

from which we obtain

$$\widehat{\Delta\boldsymbol{\theta}}(\mathcal{D}_{-i,\epsilon}) - \widehat{\Delta\boldsymbol{\theta}}(\mathcal{D}) = -\epsilon \left(\boldsymbol{J}_{F,\Delta\boldsymbol{\theta}}\big|_{\epsilon=0,\Delta\boldsymbol{\theta}=\widehat{\Delta\boldsymbol{\theta}}(\mathcal{D})}\right)^{-1} \boldsymbol{J}_{F,\epsilon}\big|_{\epsilon=0,\Delta\boldsymbol{\theta}=\widehat{\Delta\boldsymbol{\theta}}(\mathcal{D})} + O(\epsilon^2)$$

$$\approx \epsilon \left(\boldsymbol{H}(\boldsymbol{\theta}^f) + \lambda\boldsymbol{I}\right)^{-1} \left(\nabla_{\boldsymbol{\theta}} L(\boldsymbol{z}_i; \boldsymbol{\theta}^f) + \nabla_{\boldsymbol{\theta}}^2 L(\boldsymbol{z}_i; \boldsymbol{\theta}^f) \widehat{\Delta\boldsymbol{\theta}}(\mathcal{D})\right).$$

$\square$

*Proof of Corollary 2.* The result follows from (7) by substituting the Gauss-Newton Hessians $\boldsymbol{H}(\boldsymbol{\theta}^f) = \boldsymbol{G}^T \boldsymbol{V} \boldsymbol{G}$ and $\nabla_{\boldsymbol{\theta}}^2 L(\boldsymbol{z}_i; \boldsymbol{\theta}^f) = v_{ii} \boldsymbol{g}_i \boldsymbol{g}_i^T$, $\nabla_{\boldsymbol{\theta}} L(\boldsymbol{z}_i; \boldsymbol{\theta}^f) = -r_i \boldsymbol{g}_i$ from (9), and simplifying. $\qquad\square$

### B.3 Simplification of Proposition 1 in the near-stationary case

If the final parameters $\boldsymbol{\theta}^f$ are near-stationary, then the parameter change $\widehat{\Delta\boldsymbol{\theta}}(\mathcal{D})$ in (7) is small. We may thus use a reverse Taylor expansion to approximate

$$\nabla_{\boldsymbol{\theta}} L(\boldsymbol{z}_i; \boldsymbol{\theta}^f) + \nabla_{\boldsymbol{\theta}}^2 L(\boldsymbol{z}_i; \boldsymbol{\theta}^f)\widehat{\Delta\boldsymbol{\theta}}(\mathcal{D}) \approx \nabla_{\boldsymbol{\theta}} L\left(\boldsymbol{z}_i; \boldsymbol{\theta}^f + \widehat{\Delta\boldsymbol{\theta}}(\mathcal{D})\right),$$

which avoids computing the Hessian $\nabla_{\boldsymbol{\theta}}^2 L(\boldsymbol{z}_i; \boldsymbol{\theta}^f)$. Equation (7) becomes

$$\widehat{\Delta\boldsymbol{\theta}}(\mathcal{D}_{-i,\epsilon}) - \widehat{\Delta\boldsymbol{\theta}}(\mathcal{D}) \approx \epsilon \left(\boldsymbol{H}(\boldsymbol{\theta}^f) + \lambda \boldsymbol{I}\right)^{-1} \nabla_{\boldsymbol{\theta}} L\left(\boldsymbol{z}_i; \boldsymbol{\theta}^f + \widehat{\Delta\boldsymbol{\theta}}(\mathcal{D})\right), \qquad (16)$$

i.e., the gradient $\nabla_{\boldsymbol{\theta}} L(\boldsymbol{z}_i; \boldsymbol{\theta}^f)$ in the standard influence function (8) is replaced with its counterpart after further training on $\mathcal{D}$. Moreover, (15) provides an expression for $\widehat{\Delta\boldsymbol{\theta}}(\mathcal{D})$ as a Newton step from $\boldsymbol{\theta}^f$.

## C   More on Related Work

In this appendix, we discuss some more closely related works individually.

**Bae et al. (2022)** [27] examine the discrepancy between influence functions and leave-one-out retraining and decompose the discrepancy into five contributions. They show that influence functions are a much better approximation to a quantity they call the proximal Bregman response function (PBRF), which they propose as an alternative gold standard to LOO re-training. The key differences between our work and [27] are as follows: 1) We propose further training based on standard training objectives (e.g., cross-entropy or mean squared error, no proximity regularization) as a different, more general gold standard. In contrast, the PBRF involves a non-standard Bregman distance objective, chosen specifically to be closer to influence functions. In their Appendix D.2, [27] also discuss further training with and without leaving out one sample, which they refer to as "two-stage LOO re-training." However, they do not consider taking an expectation over random trainings as we do. 2) Our unified view encompasses TDA methods beyond influence functions, including more recent methods such as TRAK [19] and DataInf [20] as well as first-order methods. We evaluate the extent to which all these methods approximate further training. [27] evaluated two TDA methods, LiSSA with and without the Gauss-Newton Hessian approximation.

**Schioppa et al. (2023)** [28] discuss problematic assumptions made by influence functions and TracIn [9] and how these can be addressed. They then show that the predictive power of influence functions and TracIn is fundamentally limited by divergence in parameters as (further) training proceeds. The latter contribution is the main point of connection with our work, where we also observe the fading of predictive power over training time, especially for first-order methods. The differences in our work are: 1) We make explicit the further training gold standard that they compare to in their experiment [28, Sec. 5.2] and we further propose its expected version with respect to stochasticity in the training algorithm. 2) We conduct a larger-scale experiment than theirs in certain dimensions, notably the TDA methods evaluated, as well as averaging over the aforementioned stochasticity and using more test points and left-out training points. [28] evaluated two methods, TracIn and a Conjugate Residual method of computing influence functions.

**Basu et al. (2021)** [24] show that influence functions become worse approximators of a training example's importance on a test prediction with increasing model depth, width and if the models are trained with no weight decay. All experiments in the paper are conducted with further training (from the already trained model, not from scratch) determining the ground truth influence. This was done for efficiency reasons however, not because of an explicit restriction to the FiMO setting. Moreover, [24] do not average over further training realizations or correct for non-converged models as we do. Their study is focused on influence functions (computed using two methods, exact and LiSSA), whereas we consider other gradient-based methods as well.

**Nguyen et al. (2023)** [25] argue that a Bayesian approach is the correct way of evaluating influences of training examples on test predictions as there is a lot of variance in most TDA estimates when

we account for random initialization of the model as well as variation in batching when training. In such situations the recommendation is to consider the distribution of attribution values for a training example, or at least consider the distribution's variance, rather than simply (individual run) point estimates. Different from our work, the experiments in the paper are conducted by re-training models from scratch, and only three gradient-based TDA methods are studied, namely influence functions (IF), Grad-Dot (GD) and Grad-Cos (GC).

The seminal paper by **Koh et al. (2017)** [12] on influence functions in ML sketched a derivation similar to ours in Section 4 and Appendix B, but in less detail (in their Section 4.2 and Appendix B, less than half a page in total). In particular, our more careful derivation of the influence function in Proposition 1 keeps the $\widehat{\Delta\theta}(\mathcal{D})$ term that [12] neglect. Appendix B.1 also discusses the damping term $\lambda\boldsymbol{I}$ at greater length. [12] evaluated two influence function methods, CG and LiSSA (without the Gauss-Newton approximation).

**Zhang & Zhang (2022)** [26] analyze the effectiveness of influence functions, but only for a 2-layer ReLU network. Moreover, the theoretical analysis is done w.r.t. the neural tangent kernel (NTK) approximation of the ReLU network. They have three key findings: i) Influence functions perform better as the width of the network increases. ii) Influence functions perform poorly with weak regularization, where the regularization tries to keep the model weights as close as possible to the initial weights. Our cosine similarity results in Figure 2 and elsewhere may be a reflection of this. iii) Influence functions perform better for examples that lie in the high density region of the training distribution. While the work of [26] was mostly theoretical, they did numerically evaluate influence functions computed in two ways, using inverse Hessian-vector products and a simpler method that takes advantage of their NTK approximation.

In addition, we mention **Arnoldi influence functions** [18] as a method falling under Section 4.3. [18] proposed to use Arnoldi iteration (a.k.a. Lanczos iteration for symmetric matrices) to approximately compute the largest-magnitude eigenvalues and corresponding eigenvectors of $\boldsymbol{H}(\boldsymbol{\theta}^f)$. The method then projects gradient vectors into the subspace of top eigenvectors of $\boldsymbol{H}(\boldsymbol{\theta}^f)$. By virtue of the orthonormal basis of eigenvectors for this subspace, the matrix inversion in (8) reduces to scalar division by eigenvalues.

During the review process, a reviewer brought up works that **perform TDA by training on test points** [46], or in a similar vein, by synthesizing an image and then unlearning it [47]. The main idea of these works is to reverse the roles of training and test data. Due to symmetry, this can still estimate influence in the usual train→test direction while often reducing computational cost. These works are quite different from ours however, as the training is on test points, not further training on training points. Moreover, they do not address the FiMODA problem or a gold standard for it.

# D  Experimental Setup Details

## D.1  Data

**Choice of tabular datasets**     We followed [27] in using two regression datasets from the UCI repository [33] (CC BY 4.0 license), Concrete Compressive Strength and Energy Efficiency. We then departed from [27] in choosing two classification datasets, FICO Challenge [34] (license unknown) and Folktables [35] (MIT license), because they are at least one order of magnitude larger than the tabular datasets in [27] in terms of the number of instances and/or features.

**Data pre-processing**     For Concrete, Energy, and FICO, we split the dataset 90%-10% into training and test sets (using the scikit-learn [48] package's `train_test_split()` with `random_state=0`) and standardized the features to have zero mean and unit variance. Handling of special feature values in FICO was done as in [49]. For Folktables, we used the person-level data from year 2018 for the US state of Massachusetts. The classification task was predicting whether a person's income is above or below USD 50,000 (task `ACSIncome`). The dataset was split 75%-25% into training and test (again using `train_test_split()` with `random_state=0`), categorical features were dummy-coded, and numerical features were standardized. For CIFAR-10 (original license unclear), we used the given split into training and test sets. We employed common data augmentation techniques such as random cropping and horizontal flipping, followed by standardization using the empirical mean and standard deviation of the training set. We used a random subset of 1000 samples as validation set. For SST-2 (original license unclear), we used the given split into training, validation, and test sets. We drew

the $m = 100$ "test" instances for evaluation from the validation set because the actual test set lacks labels for computing losses. Text was tokenized using the tokenizer of the BERT model (see next subsection) with a maximum sequence length of 128 (more than sufficient for all SST-2 examples).

## D.2 Training of Final Models

**Tabular datasets** For all tabular datasets, we used a 2-hidden-layer multi-layer perceptron (MLP) with 128 units in each hidden layer. For the regression problems, the MLP has one output and the loss function is mean squared error (MSE), while for (binary) classification, the MLP has two outputs (logits) and the loss function is cross-entropy. The MLPs were trained using SGD for $T = 1000$ epochs and a batch size of 128 (following [27]) to yield the final model $f(\boldsymbol{x}, \boldsymbol{\theta}^f)$, except for the larger Folktables dataset where $T = 100$. Learning rates were as follows: 0.3 for Concrete and Energy, 0.001 for FICO, and 0.01 for Folktables.

**CIFAR-10** We used a ResNet-9 architecture [39] and trained it using SGD to minimize cross-entropy loss on the CIFAR-10 training set for $T = 50$ epochs. We trained with a batch size of 512, learning rate of 0.4, and weight decay of 0.001.

**SST-2** We used a pre-trained `bert-base-uncased` model[4] [40] (Apache 2.0 license) and attached a linear classification layer that maps the BERT model's final representation of the CLS token to logits. We fine-tuned this model on the SST-2 training set using AdamW [41] to minimize cross-entropy loss, with a batch size of 64, learning rate of $10^{-5}$, zero weight decay, and gradient norm clipping at a threshold of 1. Accuracy on the SST-2 validation set was used to choose the number of epochs from a range of 1 to 30. This resulted in a final model with validation accuracy of 92.9% after epoch 19.

## D.3 Implementation of further training

**Optimizer and parameters** For the case in which the further training algorithm $A'$ is the same as the initial training algorithm $A$, the learning rate for $A'$ was chosen to be one order of magnitude smaller than that for $A$, i.e., 0.03 for Concrete and Energy, $10^{-4}$ for FICO and Folktables, 0.04 for CIFAR-10, $10^{-6}$ for SST-2. The reason is that the given parameters $\boldsymbol{\theta}^f$ are closer to convergence than the initial parameters $\boldsymbol{\theta}^{(0)}$. The maximum number of epochs $T'$ for $A'$ was also chosen to be a fraction of $T$: $T' = 500$ for Concrete and Energy following [27], $T' = 200$ for FICO, $T' = 25$ for Folktables, $T' = 10$ for CIFAR-10, and $T' = 1$ for SST-2. Other parameters such as batch size were the same as for $A$. For the experiment on tabular data in which $A'$ was Adam, smaller learning rates were used since Adam is a more powerful optimizer: $10^{-4}$ for Concrete and Energy, $10^{-6}$ for FICO and Folktables.

**Random seeds** Regarding the random seeds (denoted as $\xi^{(s)}$ in (13), (17)) that determine the order of instances in each epoch, this control was achieved in PyTorch [50] as follows: First, the seed of a random number `Generator` object was set to $s = 0, 1, \ldots, r - 1$ ($r = 100$) using `generator.manual_seed()`. Then, a `RandomSampler` object was instantiated with this `Generator` (`RandomSampler(generator=generator, ...)`), and a `DataLoader` was instantiated with the `RandomSampler` (`DataLoader(sampler=sampler, ...)`).

**Leaving out instances** To achieve the effect of leaving out one training instance $\boldsymbol{z}_i$, we followed [28] in subtracting a second loss term $L(\boldsymbol{z}_i; \boldsymbol{\theta})/n$ from the usual loss for *every* batch. The scaling by $1/n$ (recall that $n$ is the number of training instances) approximately cancels out the weight of instance $\boldsymbol{z}_i$ in the usual loss. Denoting by $B$ and $n_B = \lceil n/B \rceil$ the batch size and number of batches per epoch, the weight of $\boldsymbol{z}_i$ in the usual loss is $1/B$ (one appearance per epoch), whereas the total weight from the subtracted loss term is $n_B/n \approx 1/B$.

**Adjustment for the effect of further training alone** As discussed in Section 6.1, we investigated two methods for adjusting for the effect of further training alone, where the results in Figure 2 use the mean subtraction adjustment in (13). For the other adjustment method in which the output due to full-dataset training is subtracted, the attribution score is given by

$$v_i = \frac{1}{r} \sum_{s=1}^{r} \left[ g\big(\boldsymbol{z}, \boldsymbol{\theta}^f + \Delta\boldsymbol{\theta}(\mathcal{D}_{-i}, \xi^{(s)})\big) - g\big(\boldsymbol{z}, \boldsymbol{\theta}^f + \Delta\boldsymbol{\theta}(\mathcal{D}, \xi^{(s)})\big) \right]. \tag{17}$$

Results corresponding to (17) are shown in Figures 5 and 9.

---

[4] `https://huggingface.co/google-bert/bert-base-uncased`

### D.4 Approximate TDA methods

All methods take the final model $f(\boldsymbol{x}; \boldsymbol{\theta}^f)$ as input, either by first computing its training loss gradients $\nabla_{\boldsymbol{\theta}} L(\boldsymbol{z}_i; \boldsymbol{\theta}^f)$ and test loss gradients, or as a model object that is used to compute derivatives.

**Grad-Dot, Grad-Cos**      Given the loss gradients, these are straightforward to implement as (normalized) inner products.

**CG, LiSSA, LiSSA-H**      We used the implementations of these methods in the `torch-influence` repository[5] (Apache 2.0 license) provided by [27]. For both CG and LiSSA, we used a damping value of $\lambda = 0.01$, which we found to yield higher cosine similarity values than the other $\lambda = 0.001$ value used by [27]. We observed that cosine similarities increase as $\lambda$ increases, so it is possible that further tuning of $\lambda$ may yield better results. For LiSSA's additional parameters, we set the `depth` (number of iterations) to 5000, the number of repeats (`repeat`) to 1, and the `scale` parameter to the smallest value in $\{10, 20, 50, 100, 200, 500\}$ for which LiSSA did not diverge.

**DataInf**      For the tabular datasets, we used the repository[6] (license unknown) provided by [20] with default parameter values, specifically `lambda_const_param=10` which determines the damping value. However for CIFAR-10 and SST-2, the authors' code was not computationally efficient enough. Instead, we implemented DataInf's key equation [20, eq. (5)] ourselves. In doing so, we approximated the average over all training instances $i = 1, \ldots, n$ with an average over a random subsample of size 1000. We also first computed the gradient inner products $L_{l,i}$, $L_{l,ik}$, $L_{l,k}$ in their eq. (5) before computing eq. (5) itself.

**EK-FAC**      We used the repository[7] (Apache 2.0 license) corresponding to [23] with default parameter settings.

**TRAK**      We used the repository[8] (MIT license) provided by [19] with default parameter settings. For the regression datasets Concrete and Energy, since the TRAK repository does not provide an implementation for regression, we tried to create one ourselves following their documentation.[9] For their "model output function," we use the model's prediction, $f(\boldsymbol{x}; \boldsymbol{\theta})$ in our notation. The squared error loss is $L(\boldsymbol{z}; \boldsymbol{\theta}) = \frac{1}{2}(f(\boldsymbol{x}; \boldsymbol{\theta}) - y)^2$, so the output-to-loss gradient that TRAK also requires is

$$\frac{\partial L(\boldsymbol{z}; \boldsymbol{\theta})}{\partial f} = f(\boldsymbol{x}; \boldsymbol{\theta}) - y.$$

### D.5 Evaluation metric

The derivations in Section 4 indicate that gradient-based methods should approximate the signed magnitudes of further training gold attribution scores. Thus, we chose an evaluation metric that is sensitive to signed magnitudes, not just rankings (evaluated for example by Spearman rank correlation, a common metric in the TDA literature). Having said this, the vectors $\boldsymbol{a}$ of gold attribution scores and $\hat{\boldsymbol{a}}$ of approximate attribution scores may have different scales. In this work, we do not concern ourselves with the differing scales, so we take both to be normalized to unit $\ell_2$ norm. In this case, the squared Euclidean distance $\|\hat{\boldsymbol{v}} - \boldsymbol{v}\|^2$ and cosine similarity $\hat{\boldsymbol{v}}^T \boldsymbol{v}$ are equivalent up to an affine transformation, and we use the latter as the evaluation metric. Note that cosine similarity is a more demanding measure than Pearson correlation (used for example in [28, 27]), since the latter corresponds to approximating each gold score $a_i$ by $\alpha \hat{a}_i + \beta$ where $\beta$ is a non-zero bias, whereas the former constrains $\beta$ to be zero. In addition, we also show results with Spearman correlation in Figure 6, which are qualitatively similar to the cosine similarity results.

### D.6 Computing resources

Experiments were run on an internal computing cluster providing nodes with 32 GB of CPU memory, V100 GPUs with 32 GB of GPU memory, and occasionally A100 GPUs with 40 or 80 GB of GPU memory. V100s were sufficient for all training however. One CPU and one GPU were used at a time.

[5] `https://github.com/alstonlo/torch-influence`
[6] `https://github.com/ykwon0407/DataInf`
[7] `https://github.com/pomonam/kronfluence`
[8] `https://github.com/MadryLab/trak`
[9] `https://trak.readthedocs.io/en/latest/modeloutput.html`

---

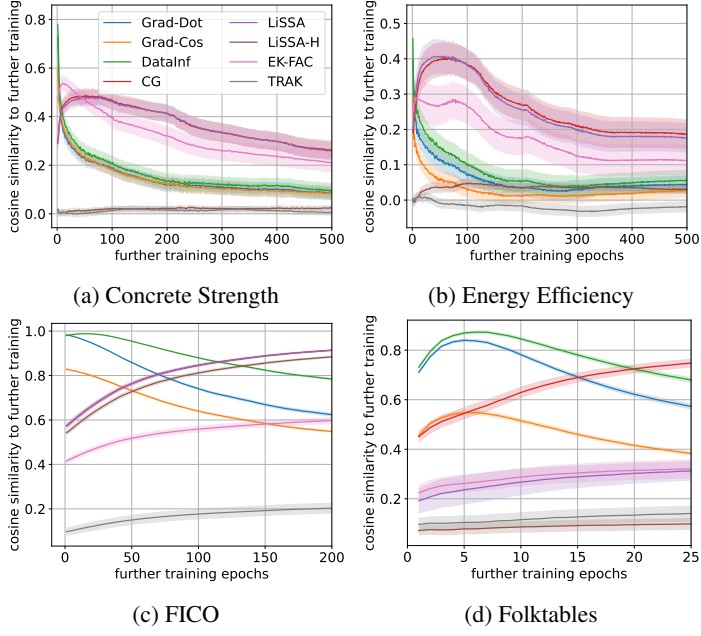

(a) Concrete Strength  (b) Energy Efficiency

(c) FICO  (d) Folktables

Figure 4: Cosine similarity between attribution scores of gradient-based TDA methods and further training as a function of the amount of further training, using a different further training algorithm (Adam) than the initial training algorithm (SGD).

The most computationally expensive part of the experiments was to perform further training on the BERT model for SST-2. Each further training run ($T' = 1$ epoch) took $\sim 12$ minutes, which includes time to evaluate intermediate checkpoints on the test set after every $\sim 10$ optimization steps. With $l = 50$ LOO instances and $r = 100$ random seeds, the total time for further training BERT was $\sim 1000$ GPU-hours. The total time for further training ResNet for CIFAR-10 was a significant fraction of this, perhaps $\sim 400$ GPU-hours. Conducting the mislabelled example detection experiment on SST-2 also required $\sim 400$ GPU-hours for further training BERT with 40 random seeds. Aside from further training, among the gradient-based methods, LiSSA was the most computationally expensive. Running LiSSA and LiSSA-H for CIFAR-10 and SST-2 took $\sim 500$ GPU-hours in total. Thus altogether, a reasonable estimate for total experiment compute is $\sim 3000$ GPU-hours.

# E  Additional Experimental Results and Discussion

## E.1  Comparison of gradient-based methods to further training

**Similarity between Grad-Dot and DataInf**  We reproduce DataInf's key equation [20, eq. (5)] below (with a sign change to be consistent with Section 4):

$$\hat{a}_k^{\text{DataInf}} = \sum_{l=1}^{L} \frac{1}{\lambda_l} \left( L_{l,k} - \frac{1}{n} \sum_{i=1}^{n} \frac{L_{l,i}}{\lambda_l + L_{l,ii}} L_{l,ik} \right), \tag{18}$$

where $k$ indexes the training instance being attributed to, $l$ the layers of the NN, and $i$ all training instances. $L_{l,i}$ and $L_{l,ik}$ are inner products between layer-specific loss gradients, as defined in [20]. The first term in (18) is like Grad-Dot (6) but with a layer-dependent weighting $1/\lambda_l$. The remaining average over $i$ can be seen as a correction term. Our results in Figures 2–5 suggest that this correction term is small.

**Different further training algorithm**  Figure 4 shows the cosine similarity between attribution scores of gradient-based methods and further training for the tabular datasets, using Adam as the further training algorithm $A'$ instead of SGD as in Figure 2. The patterns are broadly similar to those in Figure 2. However, for Folktables in Figure 4d, the curves for the first-order methods and DataInf do not peak at the beginning but rather after a few epochs. In Figures 4a and 4b, the curves for the influence function methods CG and LiSSA are also seen to peak and then decay.

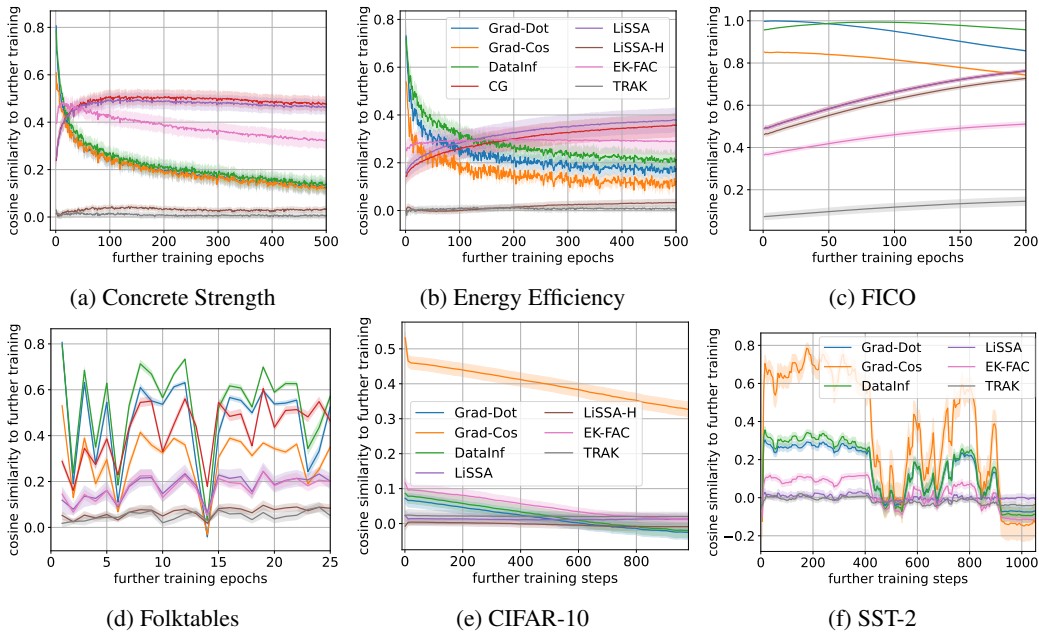

(a) Concrete Strength      (b) Energy Efficiency      (c) FICO

(d) Folktables      (e) CIFAR-10      (f) SST-2

Figure 5: Cosine similarity between attribution scores of gradient-based TDA methods and further training using the same further training algorithms as in Figure 2, but with adjustment done according to (17).

**Different adjustment for the effect of further training alone**      In Figure 5, the further training optimizer is the same as in Figure 2 ($A' = A$), but the adjustment for the effect of further training alone is done using (17), i.e., by subtracting the output from full-dataset training. The plots for Concrete, Energy, FICO, and CIFAR-10 are similar to their counterparts in Figure 2. However for Folktables and SST-2 in Figure 5d, 5f, the curves are very noisy. The reason is that the adjustment in (17) still leaves a non-negligible constant bias, i.e., the attribution scores $a_i$ are shifted up or down by a constant that varies from epoch to epoch. This varying bias is reflected in the cosine similarities shown in Figure 5d, 5f.

**Different evaluation metric**      In Figure 6, we show a counterpart to Figure 2 where cosine similarity between attribution vectors $a$ and $\hat{a}$ is replaced by Spearman correlation (and then averaged over test instances as before). The plots are qualitatively similar to those in Figure 2, particularly in terms of the grouping of the gradient-based methods. For example in Figure 6c, DataInf, Grad-Dot, and Grad-Cos remain at the top, while CG and LiSSA are next and again coincide with each other. In Figure 6d, CG rises and crosses over the top three (DataInf, Grad-Cos, Grad-Dot), similar to Figure 2d. In Figure 6f, DataInf and Grad-Dot are again highest and EK-FAC is again in third place.

**Different numbers of random seeds averaged**      In Figures 7, 8, and 9, we plot the maximum cosine similarity over epochs/steps as a function of the number of random seeds averaged. The detailed procedure is as follows. Given a number $r$ of seeds to be averaged, we divide the 100 seeds that we have in total into $100/r$ groups of $r$ seeds. For each group of $r$ seeds, we obtain a gold attribution vector $a$ using either (13) or (17) depending on the adjustment method, and compute its cosine similarity with a gradient-based attribution vector $\hat{a}$. We then compute means and standard errors of the cosine similarities over the $m$ test instances as well as the $100/r$ groups. This yields mean cosine similarities as a function of further training epochs or steps, like in Figures 2–5. Finally, we take the maximum over epochs or steps to give us the $y$-axis value in Figures 7–9 corresponding to the number of seeds $r$ on the $x$-axis. This procedure is repeated for different values of $r$.

In all cases in Figures 7–9, the cosine similarity increases with the number of seeds, implying that the averaging of further training runs brings it closer to what gradient-based methods estimate. The increases are more notable for the first-order methods and DataInf. This may appear so because the first-order-like methods can achieve higher maximum cosine similarities than second-order methods, provided there is sufficient averaging. The increases are especially notable in Figure 9

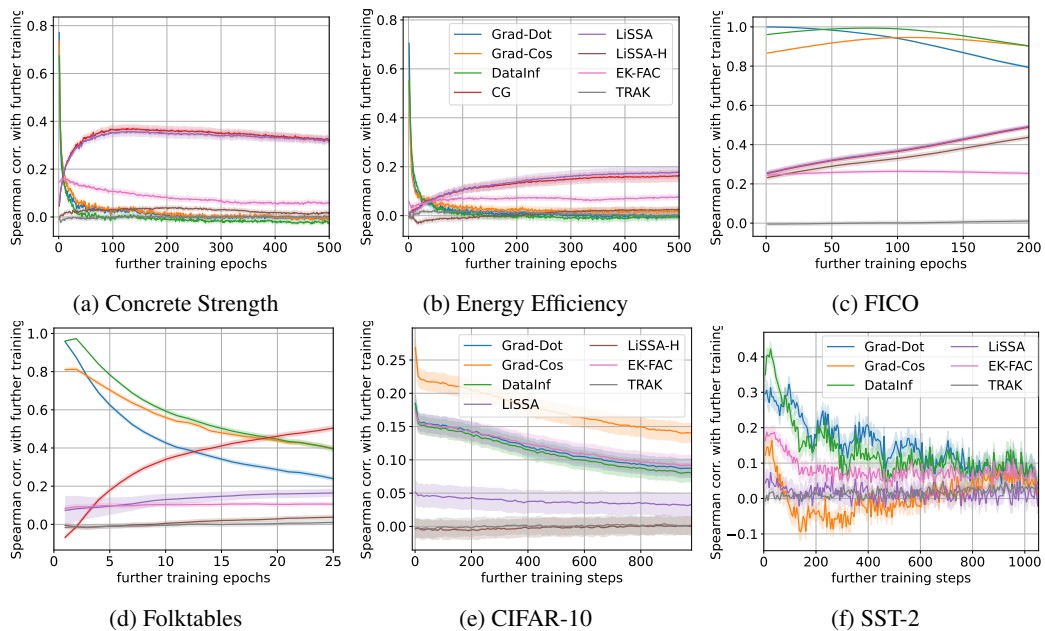

Figure 6: Spearman rank correlation between attribution scores of gradient-based TDA methods and further training, as a function of the number of epochs or steps of further training.

when adjustment is done using full-dataset further training (17). As seen in Figure 5, this adjustment method is noisier and appears to benefit more from averaging.

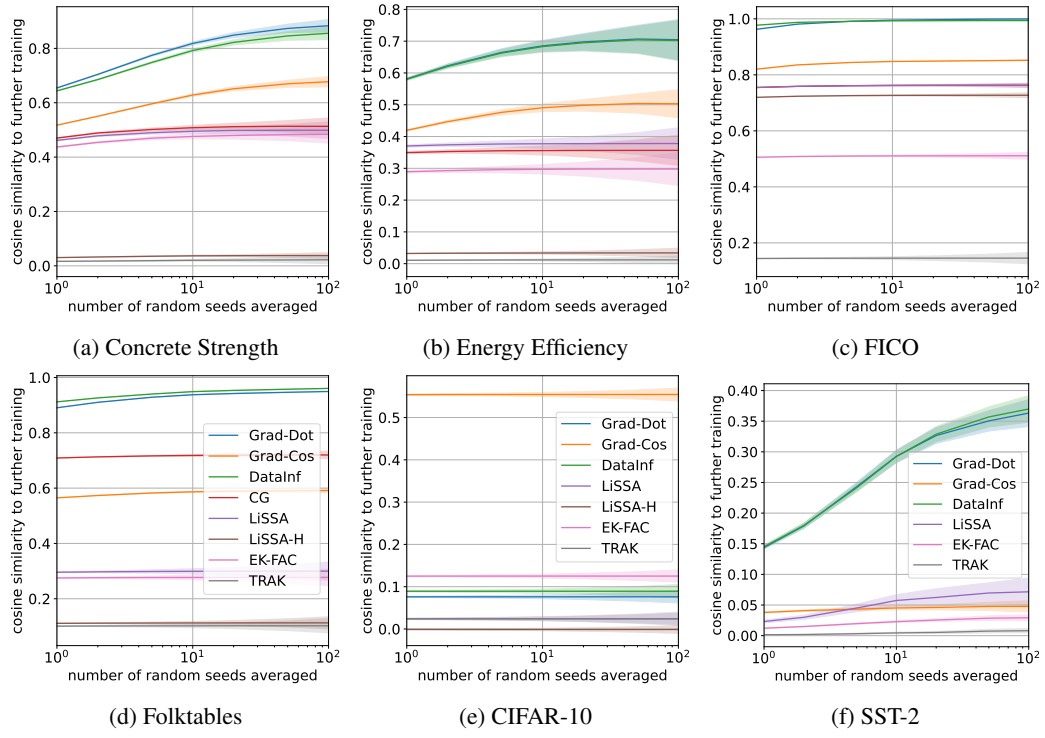

Figure 7: Maximum cosine similarity between attribution scores of gradient-based TDA methods and further training, as a function of the number of random seeds averaged. Here the further training algorithm $A'$ is the same as the initial training algorithm.

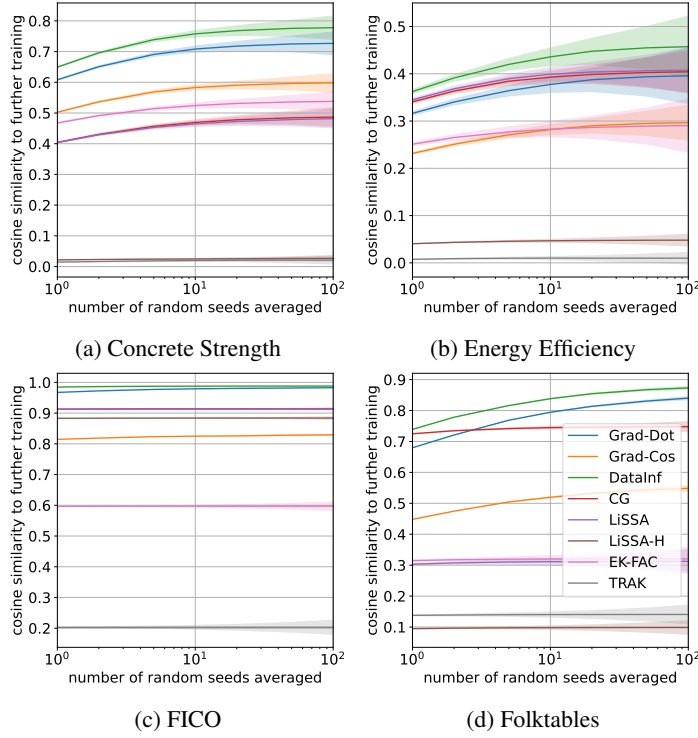

Figure 8: Maximum cosine similarity between attribution scores of gradient-based TDA methods and further training, as a function of the number of random seeds averaged. Here the further training algorithm (Adam) is different from the initial training algorithm (SGD).

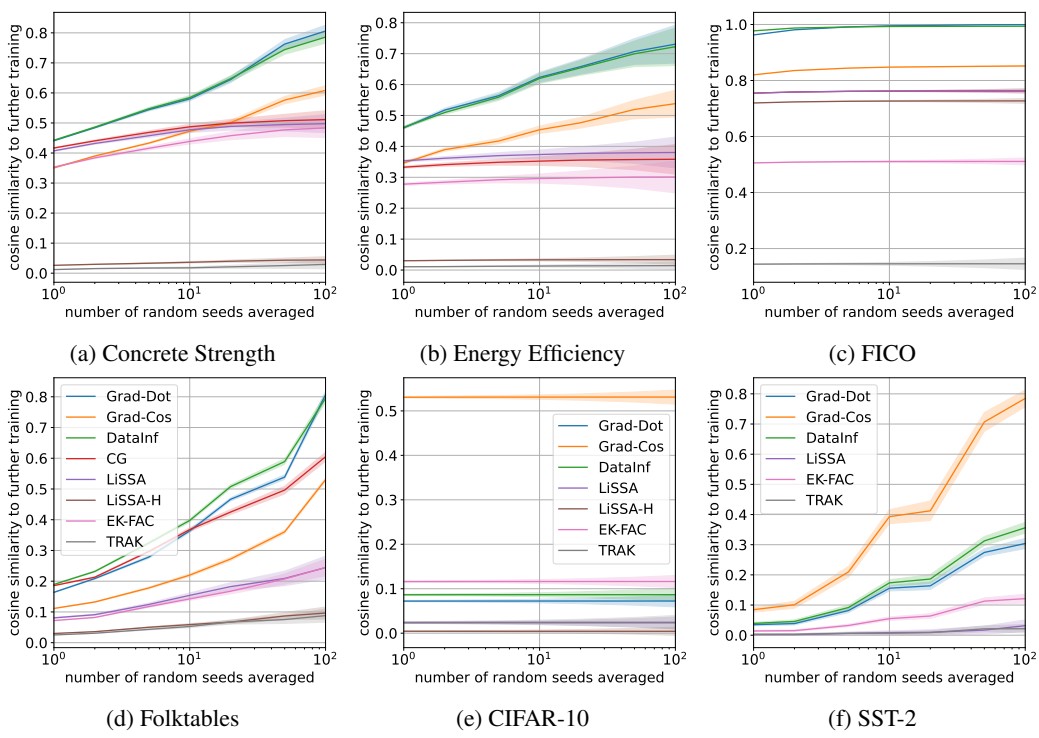

Figure 9: Maximum cosine similarity between attribution scores of gradient-based TDA methods and further training using full-dataset adjustment (17), as a function of the number of random seeds averaged.

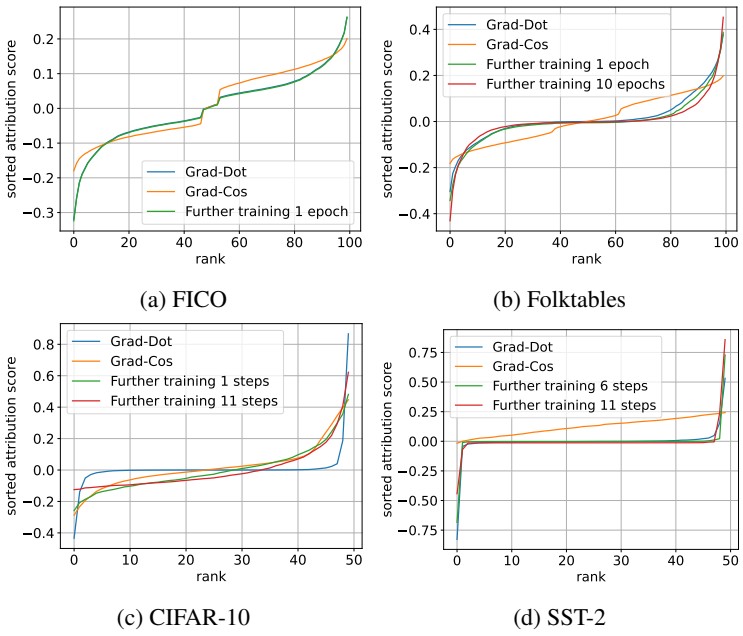

(a) FICO

(b) Folktables

(c) CIFAR-10

(d) SST-2

Figure 10: Attribution scores, sorted and averaged over test instances, from further training as well as Grad-Dot and Grad-Cos.

**TRAK$_{M=1}$** There are two possible reasons for the poor approximation quality of TRAK$_{M=1}$ seen in Figure 2 and elsewhere. First, in the FiMO setting, we have only $M = 1$ checkpoint and can only evaluate the TRAK$_{M=1}$ variant as mentioned in Section 4.3.3, i.e., without ensembling. [19] show that using an ensemble of $M \gg 1$ checkpoints greatly improves performance. Second, we compute similarity with further training attribution values, whereas [19] evaluate using their linear datamodelling score (LDS) metric, which is based on re-training and for the TAA setting.

**CIFAR-10 and SST-2** The cosine similarities for CIFAR-10 and SST-2 in Figures 2e, 2f and elsewhere are significantly lower than those for the tabular datasets. As a first step to understanding why, we plot in Figure 10 the gold attribution scores from further training (13), where for each test instance, we have sorted the scores $a_i$ in increasing order, and then averaged over the $m$ test instances. We also sort and average the attribution scores from Grad-Dot and Grad-Cos in the same manner. For FICO and Folktables in Figures 10a, 10b, the further training attribution scores are more "typical" in that they are approximately symmetric (positive versus negative) and have tails of larger positive and negative values. For CIFAR-10 however (Figure 10c), the curves are more "hockey-stick"-shaped, with a heavier positive tail and not much of a negative tail; this pattern becomes more pronounced after 11 steps of further training than after 1 step. In this case, the Grad-Cos curve offers a better match than the Grad-Dot one. For SST-2, the gold attribution score curves are angular: almost all of the scores are close to zero, with only one or two significant non-zero values at either end. In both Figures 10c and 10d, Grad-Dot and Grad-Cos struggle more to approximate these less typical attribution score curves. Further investigation into the poorer approximation quality is ongoing.

**Comparison to re-training** Figure 11 shows cosine similarities between attribution scores of gradient-based TDA methods and LOO re-training (from scratch) for the tabular regression datasets. In stark contrast to further training in Figure 2, the gradient-based attribution scores are nearly uncorrelated with LOO re-training.

## E.2 Validation of further training

### E.2.1 Mislabelled example detection

Further training is a natural choice for a gold standard because it is the analogue to re-training in the FiMO setting and is approximated by existing gradient-based methods. To further support the validity of this choice, we conducted an experiment to determine how well further training aligns with

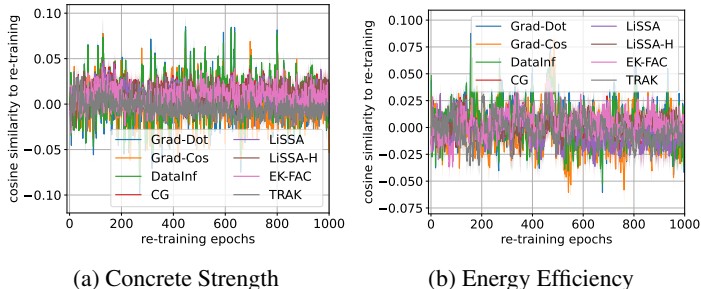

(a) Concrete Strength        (b) Energy Efficiency

Figure 11: Cosine similarity between attribution scores of gradient-based TDA methods and LOO re-training (from scratch), as a function of the number of epochs of re-training.

a downstream application of TDA, namely mislabelled data detection. In the context of FiMO and further training, we pose this as a question of whether the *sensitivity* of a final model, as measured by further training influence values, is correlated with mislabelled examples. The setup for this experiment is similar to that in Section 6.1 and prior works [20, 12]. For the binary classification datasets, we flip the labels of a randomly selected 20% of the training subset $\mathcal{L}$ on which we perform LOO further training. We then evaluate the correlation of further training influence values with the indicator of flipped instances, as measured by AUC-ROC following [20]. After one epoch of further training, the AUC values are as follows (mean and standard error over 40 trials): FICO: $0.758 \pm 0.008$; Folktables: $0.627 \pm 0.012$; SST-2: $0.995 \pm 0.003$. The last value is high because the further training influence values of flipped and non-flipped points are almost perfectly separable. We conclude that further training is indeed aligned with mislabelled data detection.

### E.2.2    Qualitative examples

The authors of the gradient-based TDA methods considered in this work have shown that examining the most influential training instances (most positive and negative) for a given test instance can give insights into what the model has learned. We refer the reader to Figure 3 of [20] and Figure 3 in the arXiv version of [19] for such examples. [23] use such influential examples to make observations about LLM phenomena, such as increasing abstraction with model scale (see their Section 5.3). In the same way that illustrative examples can be provided for approximate TDA methods, we can do the same for the further training gold standard. However, given the computational cost of further training and the experimental setup in Section 6.1, we are limited to identifying most influential examples within the evaluated subset $\mathcal{L}$ of training instances. With this limitation in mind, we discuss our findings for two of the datasets.

**Folktables**    In Table 2, we show the training instances with the most positive and most negative influence values for the first two test instances evaluated from the Folktables dataset. Since the most positive influence value implies that the loss on the test instance increases when the training instance is left out, we call this the most "helpful" training instance, and similarly call the training instance with the most negative influence value the most "harmful". Similar to previous work (for example Figure 3 in the arXiv version of [19]), the most influential training instances are semantically similar to the test instance, and the most helpful instances tend to have the same label while the most harmful have the opposite label. For example, the first test instance is a 61-year-old white man who likely has a blue-collar profession ("Sailors, Marine Oilers, Ship Engineers") and earns more than 50,000 USD a year (high income). The most helpful and harmful training instances are also white men with blue-collar jobs (construction laborer and structural metal fabricator/fitter). The most helpful instance also earns more than 50,000 and is younger at 38 years old, while the most harmful instance earns less than 50,000 and is older at 84.

**SST-2**    We also examined the most influential training instances within $\mathcal{L}$ for the $m$ test instances that we evaluated from SST-2, both individually as well as an average over the test instances. Interestingly, we find that in almost all cases, the same two training instances are most influential. These are the texts "outnumber the hits by three-to-one" (negative sentiment) and "a fresh infusion" (positive sentiment). We thus infer that these two instances were the most important within $\mathcal{L}$ for the BERT model to learn in order to classify sentiment well *in general*. We also note that "outnumber the

Table 2: Most influential training instances for the first two evaluated test instances from the Folktables dataset.

| Feature | Test instance | Most helpful training instance | Most harmful training instance |
|---|---|---|---|
| Label (income) | High | High | Low |
| Age | 61 | 38 | 84 |
| Work hours | 42 | 15 | 29 |
| Category of worker | Employee of private for-profit | Employee of private for-profit | Employee of private for-profit |
| Schooling | Associate's degree | Master's degree | No schooling completed |
| Marital status | Married | Never married | Married |
| Occupation | Sailors, Marine Oilers, Ship Engineers | Construction Laborers | Structural Metal Fabricators & Fitters |
| Place of birth | Massachusetts | New Jersey | Massachusetts |
| Sex | Male | Male | Male |
| Race | White alone | White alone | White alone |
| Label (income) | High | High | Low |
| Age | 39 | 24 | 56 |
| Work hours | 50 | 60 | 45 |
| Category of worker | Employee of private for-profit | Employee of private for-profit | Employee of private for-profit |
| Schooling | GED or alternative credential | Bachelor's degree | Some college, < 1 year |
| Marital status | Married | Never married | Married |
| Occupation | Office Clerks, General | 1st-Line Supervisors Of Food Workers | Lic. Practical & Vocational Nurses |
| Place of birth | Massachusetts | Connecticut | Massachusetts |
| Sex | Female | Male | Female |
| Race | White alone | White alone | White alone |

hits by three-to-one" is a more difficult example to classify as negative, since it requires understanding "hits" (normally positive) and "outnumber" and combining the two.

## F   Additional Limitations and Future Work

Below we summarize some additional limitations of this work that were mentioned in passing in the main text:

- **FiMO setting**: As mentioned in Section 3, a limitation of the FiMO setting is that it does not seem to permit estimation of the *contribution* of a training instance to the final model by "going back in time." It may only be possible to measure *sensitivity* to a training instance (i.e., a localized notion of contribution). In addition, a reviewer noted that sensitivity may be difficult to measure if the model has saturated on a training instance.

- **Approximation by gradient-based methods**: The fundamental assumption for gradient-based methods to approximate further training is that the amount of further training is limited, as stated at the beginning of Section 4. This limitation is evidenced by the decay of the cosine similarity curves in Figure 2 and elsewhere, especially of the first-order methods.

We elaborate on the additional points for future work listed in Section 7.

1) An open question is how much further training is the right amount to determine the final model's sensitivity to training instances. If too little, the sensitivity may not be measurable, whereas if too much, then the measurement may no longer be local, i.e., specific to the given model $f(x; \theta^f)$. The extent to which further training is well-approximated by Taylor expansions (as evaluated for example in Figure 2) may provide a partial answer.

2) We restricted attention in this work to LOO further training, as mentioned in Section 3. Future research on TDA in general could focus more on units larger than a single instance (referred to as "group influence" [14, 16, 43]), beyond making the additivity assumption that the attribution score for a group is the sum of LOO attribution scores.

3) While we have presented further training as applied to the original training set $\mathcal{D}$ and assumed that $\mathcal{D}$ is available, there is no obstacle to considering further training on new, unseen instances. This would broaden the problem to predicting the effect of further training on new data and connect to problems such as data selection.

# G   Broader Impacts

This paper highlights the final-model-only setting for TDA, which is a setting that allows a wider set of practitioners (for example HuggingFace model users) to perform TDA because it does not assume access to the training algorithm or intermediate information from training. At the same time, this wider access to TDA might lead to increased misinterpretation or misplaced trust in attribution scores, especially among less expert users. Modern neural networks and neural network training procedures are highly complex, and any insights obtained into them should be tempered with some scientific skepticism. Beyond these considerations, there may be other potential societal consequences of our work, in line with machine learning research in general. Hence we do not feel that they must be specifically highlighted here.

