# OpenReview forum: "Final-Model-Only Data Attribution with a Unifying View of Gradient-Based Methods"
_NeurIPS.cc/2025/Conference — NeurIPS 2025 poster_

### Official Review · Reviewer_hy33 · 2025-06-26

**Clarity:** 3
**Significance:** 2
**Originality:** 1
**Rating:** 3
**Confidence:** 4

**Summary:**

This paper addresses the Final-Model-Only (FiMO) setting for training data attribution (TDA), in which only a trained model’s weights are available. The authors introduce a gold-standard based on “further-training” sensitivity, which trains the final checkpoint on (a) the full dataset and (b) the dataset with one example removed, then measures the resulting change in a test loss or prediction, averaged over multiple random restarts. The author show that this gold-standard recovers first-order and second-order TDA scores. Empirically, they compare these classic TDA methods against the gold-standard on different benchmarks.

**Questions:**

- Apart from cosine similarity, could you also consider correlation coefficient or other measure to give us a more comprehensive view comparing between further training and those gradient-based baselines?
- Just to make sure, is the very original IF already within the so-called FiMO setting?
- For the low quality of the gradient-based approximation, it can just because the approximation is not good. Under this setting, we might also need to assume the further training objective is not deviating form the original one too far, which is usually hard to justify in real world. Will it be interesting to just "come up with a good enough further training obj" for better attribution?
- How does the damping term in second-order method affect the quality of attribution?
- Is there any experiment on generation settings?
- Conceptually, how different it is between your further training and the "two-stage retraining" introduced in [1]?



[1] Bae, Juhan, et al. "If influence functions are the answer, then what is the question?." Advances in Neural Information Processing Systems 35 (2022): 17953-17967.

**Ethical Concerns:**

["NO or VERY MINOR ethics concerns only"]

**Final Justification:**

During rebuttal, the authors resolve most of my concerns. However, I feel the empirical results, the relation between gradient-based methods and further training could not justify to be as the "golden-standard" for the setting. (echoing reviewer NTqP) This makes me hesitate to lean toward acceptance.

**Limitations:**

yes

**Quality:**

2

**Strengths And Weaknesses:**

## Strengths

- Theoretical unification:
Shows that Influence Functions, TracIn, gradient‐dot/cos, etc., serve as Taylor approximations of further training.

- Novel scenario:
The paper claims they are the first to consider the FiMO setting. And the golden standard, which is the further training, makes sense.

## Weakness

- The training and testing data size seem to be too small to draw insightful and consistent conclusion.
- Although the setting is claimed to be new, it is actually just a variation of saying "re-training is so expensive and we can't compute gradients". So literature like [1] implicitly already discusses similar situation. There is also work [4] utilizing "unlearning" as TDA, which also resemble your core idea.
- In terms of further training, works like [2] also discuss the possibility of "further LoRA fine-tuning as TDA". Although they did use gradient information, the idea is similar in some sense. The ensemble idea they propose also match the expected value notation in equation (3).
- Looking at Figure 2, I will be hesitate to claim the cosine similarity between these baseline TDA methods and their further training counterparts are actually similar...Especially the setting is more complex like (e)(f). The curves in Figure 5 are even harder to explain.



[1] Ko, Myeongseob, et al. "The mirrored influence hypothesis: Efficient data influence estimation by harnessing forward passes."
Proceedings of the IEEE/CVF Conference on Computer Vision and Pattern Recognition. 2024.

[2] Deng, Junwei, et al. "Efficient ensembles improve training data attribution." arXiv preprint arXiv:2405.17293 (2024).

[3] Park, Sung Min, et al. "Trak: Attributing model behavior at scale." arXiv preprint arXiv:2303.14186 (2023).

[4] Wang, Sheng-Yu, et al. "Data attribution for text-to-image models by unlearning synthesized images." Advances in Neural Information Processing Systems 37 (2024): 4235-4266.

[5] Choi, Woosung, et al. "Large-Scale Training Data Attribution for Music Generative Models via Unlearning." arXiv preprint arXiv:2506.18312 (2025).

---

> ### Author Rebuttal · Authors · 2025-07-30
>
> Thank you very much for your review. It appears from your originality score of 1 that originality is your main concern. As we discuss below, we think that the **works you cited** to support this assessment **are actually complementary to our contributions** rather than similar to them. We would be happy to discuss these works further with you. We also think that the last weakness you identified **(lower cosine similarities) is not a weakness of our paper**, and we respond to your concern about data size.
>
> ### Setting
> > Although the setting is claimed to be new, it is actually just a variation of saying "re-training is so expensive and we can't compute gradients".
>
> To clarify possible misunderstanding: We do assume it is possible to compute gradients of training instances, as otherwise (gradient-based) further training and the gradient-based approximations to it would not be possible. And the motivation for considering the FiMO setting is not just that re-training is expensive. We are considering the scenario where the original training process is *inaccessible* altogether (e.g., a model downloaded from Hugging Face). Given this constraint, our main contributions are as summarized in lines 51-54 of the introduction: We formulate an alternative gold standard that also operates under this constraint and show the relationships of existing gradient-based methods to this gold standard.
>
> ### Works that you cited
>
> > So literature like [1] implicitly already discusses similar situation. There is also work [4] utilizing "unlearning" as TDA, which also resemble your core idea.
>
> The main idea of both [1] and [4] is to reverse the roles of training and test data and use the effect on a *training* point of training on a *test* point [1], or unlearning a synthesized image [4], to approximate influence in the opposite direction (the usual one). While a valuable idea, we see it as complementary to our contributions summarized above. Specifically, this idea could be applied to yield reversed versions of the gradient-based methods that we study, but it does not address the question of an alternative gold standard for the FiMO setting or its relationships with gradient-based methods.
>
> > In terms of further training, works like [2] also discuss the possibility of "further LoRA fine-tuning as TDA". ... The ensemble idea they propose also match the expected value notation in equation (3).
>
> We also think that [2] is complementary to our main contributions. We mentioned in Appendix F, lines 1219-1221 that LoRA fine-tuning could be used to make the further training gold standard more efficient, and this could be done using the specific method of [2], which we are happy to cite for this purpose. However, [2] does not address the question of a gold standard for the FiMO setting or its relationships with gradient-based methods. In addition, while the idea of ensembling has been proposed to improve practical TDA methods (for example TRAK by Park et al. (2023), which predates [2]), our work is the first to also propose it for a gold standard itself, to make it more reliable.
>
> ### Lower cosine similarities
>
> > Looking at Figure 2, I will be hesitate to claim the cosine similarity between these baseline TDA methods and their further training counterparts are actually similar...Especially the setting is more complex like (e)(f). The curves in Figure 5 are even harder to explain.
>
> - We agree with you (see point 1 in lines 377-378 and its elaboration in Appendix F). We would only say that the first-order-like methods (Grad-Dot, Grad-Cos, DataInf) are “actually similar” to further training when the amount of further training is low and the cosine similarity is high enough, and perhaps not at all in Figure 2(e)(f). Figure 5 looks less pleasing because it is an ablation showing that an alternative adjustment method (see lines 303-306 and Appendix D.3) works less well.
> - In any case, we do not see these lower cosine similarities as a weakness of our paper because we are not advocating for a particular TDA method and are merely reporting how well current methods approximate further training. From this perspective, we might say that they reflect weakness of the current methods and open up possibilities for future research.
>
> ### Data size
>
> > The training and testing data size seem to be too small to draw insightful and consistent conclusion.
>
> - First, we used the full training set for further training.
> - Second, we evaluated influence scores only for a subset of training instances ($|\mathcal{L}| =$ 50 or 100) because the purpose of the experiment was to compare to the further training gold standard. The latter constrains $|\mathcal{L}|$ because the number of further trainings is proportional to $|\mathcal{L}|$. Moreover, 50 or 100 training instances compares favorably to previous retrospective work on influence functions (Bae et al. (2022) used 20 training instances (see their Appendix C.2), Schioppa et al. (2023) used 32 in their Section 5.2).
> - Third, regarding test instances, we evaluated 100 to assess variation across test instances; this number also compares favorably to Schioppa et al. (2023) who used 16. Figure 2 and similar figures show that the error bars resulting from 100 test instances are reasonable and our conclusions are reliable.
>
> ### Your questions
>
> > Apart from cosine similarity, could you also consider correlation coefficient or other measure to give us a more comprehensive view comparing between further training and those gradient-based baselines?
>
> Indeed, we use Spearman correlation as an alternative measure in Appendix E.1, Figure 6, with qualitatively similar results.
>
> > Just to make sure, is the very original IF already within the so-called FiMO setting?
>
> Yes, and one of the points we are making is that influence functions are more suited to the FiMO setting than to the full retraining setting.
>
> > For the low quality of the gradient-based approximation, it can just because the approximation is not good. Under this setting, we might also need to assume the further training objective is not deviating form the original one too far, which is usually hard to justify in real world. Will it be interesting to just "come up with a good enough further training obj" for better attribution?
>
> The problem is not that the further training objective deviates from the original training objective, since we have shown that the gradient-based methods approximate further training, and we compare them only to further training, not original training. We think it may be interesting to further investigate how the further training *algorithm* affects the quality of this approximation. We touch on this in Appendix E.1, Figure 4, where we used Adam as the optimizer instead of SGD.
>
> > How does the damping term in second-order method affect the quality of attribution?
>
> For the second-order methods CG and LiSSA, which use a damping parameter $\lambda$, please see Appendix D.4, line 1042 and onward for discussion of our choice of $\lambda$. We observed that cosine similarities increase with $\lambda$.
>
> > Conceptually, how different it is between your further training and the "two-stage retraining" introduced in [1]?
>
> We mention in Appendix C, lines 913-915 that the main difference is our proposal of averaging to mitigate the randomness of training.

---

### Official Review · Reviewer_NTqP · 2025-06-28

**Clarity:** 3
**Significance:** 1
**Originality:** 2
**Rating:** 3
**Confidence:** 4

**Summary:**

This paper proposes "further training" as a new evaluation measure (or golden standard) for training data attribution (TDA) methods in the "final-model-only" (FiMO) setting, where practitioners only have access to trained model weights rather than the full training process (e.g., intermediate checkpoints or training algorithm). The authors justify the proposed metric by showing that many existing TDA methods can be seen as approximations to "further training" through Taylor expansions around the final checkpoint. Across tabular, image, and text tasks, the authors demonstrate interesting trade-offs between different TDA methods: first-order methods (Grad-Dot, Grad-Cos) achieve high initial approximation quality but decay with increased further training, while second-order influence function methods (CG, LiSSA, EK-FAC) maintain more stable approximation quality over training steps but often underperform compared to first-order methods.

**Questions:**

Please see the weakness section above for my questions and suggestions. The most unconvincing aspect of this paper is that the conclusions drawn from further training differ significantly from those obtained when methods are evaluated under retraining-based approaches. Given this discrepancy, I am not convinced that further training should be treated as a gold standard for data attribution.

**Ethical Concerns:**

["NO or VERY MINOR ethics concerns only"]

**Final Justification:**

As I mentioned in the rebuttal, I find the framing of calling this tool the golden standard for TDA (even at FiMO setting) problematic. I think the manuscript requires a significant modification to address this concern.

**Limitations:**

Yes

**Quality:**

2

**Strengths And Weaknesses:**

Strengths:
- The paper is well-motivated and clearly written with a detailed problem formulation. All derivations in the main manuscript appear correct (I haven't read the appendix in detail, but skimmed through).
- Experiments cover a wide range of tasks, and a detailed experimental setup is provided in the appendix.

Weaknesses:
- Although the authors clearly articulate the rationale for categorizing TDA into three settings (which I agree with), I don't agree that different settings require different gold standards. There are classes of algorithms more applicable to each setting (e.g., MAGIC [1] for TAA, TraceIn [2] for CPA, and Grad-Dot for FiMO), but the fundamental objective of data attribution stays the same: understanding which training examples led to particular model outputs. The natural ground truth for all data attribution should be retraining-based approaches. While "further training" can serve as a proxy metric (just like mislabeled data detection), TDA algorithms should ultimately be evaluated using retraining-based metrics. I am not convinced that "further training" should be treated as the gold standard in any setting. Why can't we evaluate different TDA algorithms in settings we have access to the entire training procedure, and use the SOTA method in TAA setup?
- On a related note, it has been widely shown in the literature that influence function based approaches outperform first-order methods on several retraining-based evaluations [3, 4, 5]. If conclusions based on "further training" differ from retraining-based metrics, wouldn't this suggest "further training" might be capturing something other than what we want from data attribution?
- Several experimental details require clarification. For instance, when leaving out a data point (line 1024), the authors subtract the loss on that point in each gradient update rather than removing it from the dataset entirely—this is quite different from further training without a data point and needs justification (or at least mention) in the main text. Additionally, second-order methods approximate further training under fixed regularization (as in the derivations authors provided), while further training is done without regularization.

Minor:
- Many important details are moved to the appendix without proper discussion in the main text. For example, the choice of cosine similarity as the evaluation metric (line 317) is not intuitive and needs explanation in the main text.
- I'm not sure what the authors meant in line 865. Further training could change the solution even when at the model is optimal under a fixed dataset. The gradient of the empirical loss being 0 does not imply the gradients is 0 for all individual data points.

[1] Ilyas, A., & Engstrom, L. (2025). MAGIC: Near-Optimal Data Attribution for Deep Learning. arXiv preprint arXiv:2504.16430.
[2] Pruthi, Garima, et al. "Estimating training data influence by tracing gradient descent." Advances in Neural Information Processing Systems 33 (2020): 19920-19930.
[3] Park, Sung Min, et al. "Trak: Attributing model behavior at scale." arXiv preprint arXiv:2303.14186 (2023).
[4] Bae, Juhan, et al. "Training data attribution via approximate unrolled differentiation." arXiv preprint arXiv:2405.12186 (2024).
[5] Deng, Junwei, et al. "Efficient ensembles improve training data attribution." arXiv preprint arXiv:2405.17293 (2024).

---

> ### Author Rebuttal · Authors · 2025-07-30
>
> Thank you very much for your review. We respect your strong position on what TDA can encompass, what the objective can be, and therefore what the gold standard should be. We will say however that if one takes the unavailability of the training procedure as an unchangeable fact in the FiMO setting (as is usually the case with a model downloaded from Hugging Face), then we think that one should at least consider an alternative objective that also respects this constraint (i.e., quantifying sensitivity rather than contribution), and accordingly an alternative gold standard.
>
> We do want to ask you the following: **Could your concern be allayed if we used different terms to distinguish our setting from the standard TDA problem?** For example, we could always qualify TDA as FiMO-TDA from the end of Section 2 onward, and we could refer to further training as a “silver standard” rather than a gold standard.
>
> For your other comments, we think that they can be largely addressed by moving content from the appendix to the main paper (as allowed for camera-ready papers), as detailed below.
>
> > when leaving out a data point (line 1024), the authors subtract the loss on that point in each gradient update rather than removing it from the dataset entirely—this is quite different from further training without a data point and needs justification (or at least mention) in the main text.
>
> We are happy to move the referenced paragraph in Appendix D.3 to the main text. As mentioned there, one justification is that [28] used a similar procedure (but they did not actually justify it). Another reason is that it seems to also mitigate the randomness of stochastic mini-batch training, where the mini-batch that a data point appears in/is left out of is random. (This could have been the reason for [28] as well.)
>
> > second-order methods approximate further training under fixed regularization (as in the derivations authors provided), while further training is done without regularization.
>
> As mentioned in line 125, we wanted further training to be like typical neural network training, so we did not apply regularization. For the second-order methods and specifically CG and LiSSA, which use a regularization/damping parameter $\lambda$, we explored varying $\lambda$ to improve the quality of approximation to further training. As mentioned in Appendix D.4, lines 1042 and onward, we found that a larger value of $\lambda = 0.01$ was better than $\lambda = 0.001$ used in prior work. We can mention this in the main text as well.
>
> > the choice of cosine similarity as the evaluation metric (line 317) is not intuitive and needs explanation in the main text.
>
> We are happy to move Appendix D.5 to the main text to elaborate on cosine similarity.
>
> > I'm not sure what the authors meant in line 865. Further training could change the solution even when at the model is optimal under a fixed dataset. The gradient of the empirical loss being 0 does not imply the gradients is 0 for all individual data points.
>
> In the convex case of Appendix A, the gradient of the empirical loss being 0 is a sufficient condition for optimality, and thus $\Delta\theta = 0$ is an optimal solution to (1) when $\theta^f = \theta^*$ is optimal. You are right that the gradient is likely non-zero at each individual data point, and this might be an issue with stochastic optimization algorithms, but this is more an artifact of that class of algorithms. Here we are making more of a theoretical statement.

---

> > ### Comment · Reviewer_NTqP · 2025-08-02
> >
> > I thank the authors for their detailed response. As the authors mentioned, pulling up relevant content from the appendix would improve the quality of the paper. Given that these issues will be addressed in the updated manuscript, I will raise my score to 3.
> >
> > However, my main concern regarding the validity of using this framework as a TDA gold standard (even in the FiMO setup) remains. I agree with the authors that tools for validating TDA method accuracy when only the final model is available (FiMO setting) would be extremely valuable to the community. My main point is this: if the findings from this paper directly contradict what we observe from standard leave-some-out experiments (e.g., LDS) supported by extensive literature, can we trust this as a valid TDA evaluation technique? I would be more comfortable calling this a silver standard if it showed even weak correlation with the actual metrics we care about in TDA, but that does not appear to be the case. For example, method X might perform best in this FiMO evaluation, but when one actually obtains the training algorithm and computes the LDS (or any retraining-based evaluation), it might perform worst.
> >
> > One way to tackle this discrepancy would be to reframe the problem entirely: introduce a new task focused on predicting the effect of leaving one data point out during further training (rather than framing this as TDA), connect that existing TDA methods can serve as useful estimation tools for this task, and demonstrate that first-order methods perform best in such scenarios with several justifications you mentioned in the paper and the rebuttal. However, this would require significant changes to the manuscript.

---

> ### Author Response · Authors · 2025-08-03
>
> Dear Reviewer NTqP,
>
> Thank you very much for thoughtfully considering our rebuttal and being wiling to raise your score.
>
> We understand your main concern. In the first part of Section 3 (to the end of page 3), we believe we are indeed reframing the problem, as one of quantifying "sensitivity" of the model and "predicting the effect of leaving one data point out during further training." Thus, we proposed in our rebuttal above to use the term "FiMO-TDA" to further distinguish our problem setting from standard TDA. We do not think that this would require significant changes to the manuscript. It would require replacements and single-sentence clarifications where the terms "TDA" and "FiMO" appear, as follows:
> - To the last paragraph of Section 2, we will add the sentence "We will use the term 'FiMO-TDA' to refer to TDA in the FiMO setting and to clearly distinguish it from the standard TDA problem." Note that the title of Section 3 is already "Final-Model-Only TDA."
> - First sentence of Section 3 will be changed to "We first consider the question of how FiMO-TDA should ideally be done, i.e., what could serve as a “gold standard” method, where the gold standard also respects the FiMO constraint." The acronym "FiMO-TDA" and phrase "where the gold standard also respects the FiMO constraint" will also be added to the similar sentence in the introduction, lines 28-29.
> - Line 94: "TDA" --> "FiMO-TDA"
> - Line 109: "the FiMO setting" --> "FiMO-TDA"
> - Line 159: "the FiMO setting" --> "FiMO-TDA"
> - Line 238: "Under the FiMO setting" --> "For FiMO-TDA"
> - Line 277: "in the FiMO setting" --> "for FiMO-TDA"
> - Line 310: "TDA" --> "FiMO-TDA"
> - Line 373: Add "(FiMO-TDA)" after "TDA"
> - Line 376: "FiMO" --> "FiMO-TDA"
> - Line 378: "TDA" --> "FiMO-TDA"
>
> We hope that, if you could be so kind as to re-read the first part of Section 3 and review the above changes, that they can address your concern.

---

> ### Comment · Reviewer_NTqP · 2025-08-06
>
> I thank the author for their comment. While I agree that addressing these issues is a step towards the right direction, I have reviewed Sections 1-3 again and believe the proposed changes are insufficient to address my concerns. Therefore, I would like to maintain my current score. I continue to have concerns about the use of "TDA" (even within the FiMO framework) for the reasons I outlined above, as I believe it may be misleading in this context. Instead of using the term TDA, I think that the work should be formulated directly in terms of predicting the effect of further training. I don't think further training can be the golden standard of TDA in any setting. If something is to be the gold standard, it should show a correlation with retraining-based metrics (e.g., actually removing the data point(s) and retraining the model from scratch).

---

> > ### Author Response · Authors · 2025-08-08
> >
> > Thank you Reviewer NTqP for the additional reply. We respect your position on what TDA can encompass (as we wrote above).
> >
> > For the record, we would like to express our preference for continuing to use the term "TDA" because of our paper's strong connections to the TDA literature. The proposed further training gold standard is closely related to the re-training from scratch gold standard for the standard TDA problem, and the gradient-based methods that we re-derive and evaluate are all from the TDA literature. In addition to the specific minor changes that we proposed in our previous reply above, we will make clearer that we are reframing the problem of TDA in the FiMO setting.

---

### Official Review · Reviewer_3d7B · 2025-07-03

**Clarity:** 3
**Significance:** 2
**Originality:** 3
**Rating:** 5
**Confidence:** 3

**Summary:**

The work is focused on the problem of training data attribution where only final trained models are considered for analysis. To measure the sensitivity of the models, further training with appropriate adjustment and averaging are proposed. Gradient-based methods are unified for TDA to show that all these methods approximate the further training gold standard in different ways. This is done through first and second-order Taylor expansions around $\theta_f$. The problem is set clearly to create notations and build onto the attribution/influence scores $a_i$ to be assigned to the training instance $z_i$. Three problem settings are considered: a) Training Algorithm Available (TAA) -- access to the training algorithm A, b) Checkpoints Available(CPA) -- no access to A but access to intermediate checkpoints available and c) Final Model Only (FiMO) -- only access to final model. This work focuses on FiMO. This is the main contribution -- providing a further training gold standard for FiMO for the leave-one-out (LOO) case.  Particularly, issues of non-convergence (tackled by adjusted computation of sensitivity to the presence of example $z_i$ -- Figure 1) and stochasticity (tackled by averaging over seeds) have been mentioned. The gradient-based methods are then unified as approximate further training, first-order methods (GradDot), Influence function methods and Gauss-Newton approximate hessian based methods are unified. Relationships with existing influence function methods are also discussed. Finally, experiments are conducted to produce two main insights -- a) approximation quality of first-order methods decays with the amount of further training and b) approximations given by influence function based approaches are more stable but lower in quality.

**Questions:**

- [**Minor Comments**]: From line 87-90, it's argued that FiMO is applicable to scenarios with TAA and CPA settings. While, this is true, the increased access in these cases may be able to provide deeper insights?  The lines 87-88 are reasonable but there could be a significant number of cases where TAA and CPA are available. The details on some limitations of FiMO analysis that can be overcome in TAA or CPA settings could perhaps be mentioned briefly.
- [**Number of Random Seeds**]: Figures 3(b) and 7(f)  show the impact of number of random seeds averaged on the maximum cosine similarity for the SST-2 dataset -- the general suggestion is that a lower number of random seeds (10-20) can be used as the highest impact is seen there. However, in these cases, DataInf seems to be benefitting from multiple seeds. This is explained in the Appendix. However, this adds to the computations. A clarification on the following two points will be useful -- (i) why DataInf is uniquely sensitive, and (ii) whether the steadily rising curve is worth the extra compute cost.
- [**Pseudocode/Implementation**]: If there are some institutional constraints on sharing the code, it may be good to provide some example pseudocode for reproducibility.

**Ethical Concerns:**

["NO or VERY MINOR ethics concerns only"]

**Final Justification:**

Upon considering the comprehensiveness of the work and detailed discussions with the reviewers, I have a positive outlook of the paper.

**Limitations:**

yes

**Paper Formatting Concerns:**

No paper formatting issues are visible.

**Quality:**

3

**Strengths And Weaknesses:**

- Strengths
  - [**Positioning**]: The paper is positioned well amongst the TDA literature and provides contextual explanations of it's proposed approaches placing them precisely in the current literature.
  - [**Unification**]: From a reader's perspective, the manuscript does a good job of providing a unifying view of gradient based methods commonly used in the influence literature.
  - [**Analyses**]: Extensive analysis is conducted to support the theoretical studies. This study provides the first practical FiMO benchmark.

- Weakness
  - [**Computational Time**]: The experiments took about 3000 GPU hours (Appendix E) on (relatively) small CNN and BERT models. The Appendix F mentions this and suggestions are made to reduce the number of seeds or using parameter efficient approaches like LoRA. However, the effects of LoRA have been left for future work and that would still not address the problem of computational time for widespread and meaningful adoption.
  - [**Scope**]: The empirical studies on non-tabular datasets have low cosine similarity on all methods. The findings on tabular datasets are not reflected in the non-tabular datasets. For instance, Figure (e) CIFAR-10 shows significantly higher cosine similarity for Grad-Cos over Grad-Dot as opposed to the findings on tabular datasets. It's mentioned that the gold attribution vectors $a$ behave differently for non-tabular datasets. Lines 1140-1153 in Appendix E.1 discuss this issue but this poor approximation quality is still a weakness in non-tabular domain.

---

> ### Author Rebuttal · Authors · 2025-07-30
>
> Thank you very much for your review and for recognizing the value of our positioning/organizing the TDA literature and unifying gradient-based methods within it. Below we first discuss why the **low cosine similarities are not a weakness of our paper**. We then respond to your concern about computation and your other questions.
>
> > **[Scope]**: The empirical studies on non-tabular datasets have low cosine similarity on all methods. … Lines 1140-1153 in Appendix E.1 discuss this issue but this poor approximation quality is still a weakness in non-tabular domain.
>
> We agree that the low cosine similarities on non-tabular datasets are a less desirable finding for researchers in this area. From the perspective of approximating further training, we think they reflect weakness of the current methods and motivate the development of higher-quality methods (see point 1 in lines 377-378 and its elaboration in Appendix F). However, we do not see this as a weakness of our paper because we are not advocating for a particular TDA method and are merely reporting how well current methods approximate further training.
>
> > **[Computational Time]**: The experiments took about 3000 GPU hours (Appendix E) on (relatively) small CNN and BERT models. The Appendix F mentions this and suggestions are made to reduce the number of seeds or using parameter efficient approaches like LoRA. However, the effects of LoRA have been left for future work and that would still not address the problem of computational time for widespread and meaningful adoption.
>
> As discussed in Appendix D.6, most of the computational time was needed because we aimed to implement and compare to the further training gold standard. Our focus in this work was to establish this gold standard and as such, we did not intend it to be computationally efficient, nor are we proposing it as a practical TDA method. The suggestions given in Appendix F (and work cited by Reviewer hy33) could ease implementation of further training in future work.
>
> > **[Minor Comments]**: From line 87-90, it's argued that FiMO is applicable to scenarios with TAA and CPA settings. While, this is true, the increased access in these cases may be able to provide deeper insights?
>
> Yes of course, if increased access is available, then we encourage practitioners to take advantage of it if they can. We can clarify that while FiMO is applicable to these settings with greater access, it may not be the most recommended.
>
> > **[Number of Random Seeds]**: Figures 3(b) and 7(f) show the impact of number of random seeds averaged on the maximum cosine similarity for the SST-2 dataset -- the general suggestion is that a lower number of random seeds (10-20) can be used as the highest impact is seen there. However, in these cases, DataInf seems to be benefitting from multiple seeds. This is explained in the Appendix. However, this adds to the computations. A clarification on the following two points will be useful -- (i) why DataInf is uniquely sensitive, and (ii) whether the steadily rising curve is worth the extra compute cost.
>
> 1. DataInf is not uniquely sensitive. The Grad-Dot curve almost coincides with it. We remark in lines 1130-1131 that the first-order-like methods (including DataInf) are generally more sensitive to the number of seeds. It might appear this way because these methods can achieve higher maximum cosine similarities than second-order methods, but it requires more averaging of further training runs to make this apparent.
> 1. Whether the extra computational cost is worthwhile is user-dependent and not something we can answer in general. It does appear that most of the gain is achieved at ~20 seeds (this might be more apparent if we plotted on a linear scale).
>
> > **[Pseudocode/Implementation]**: If there are some institutional constraints on sharing the code, it may be good to provide some example pseudocode for reproducibility.
>
> We will revisit the question of sharing code with our institution and we are hopeful that we will be able to do so for the purpose of reproducibility. In particular, we hope to be able to share our specific code for further training and the indices of the left-out training instances in $\mathcal{L}$.

---

> > ### Comment · Reviewer_3d7B · 2025-08-05
> >
> > I thank the authors for their to the point responses.
> > The contributions are towards showing how current methods in TDA literature approximate further training. The low cosine similarity on non-tabular datasets indicate that these methods have a poor approximation. These insights are useful but need to be more pronounced in the text. In essence, I understand the contribution should not be judged on whether the reported metrics are higher or lower. However, since the core contribution is focused on unification, it would be good to provide a section on the core insights obtain overall from the paper.  The results section does a good job of  explaining the results from the experiments supported by appendix where the experimental setup if further detailed. The analysis should be summarised into core discussion points and how/why these are similar or different from the findings in the literature (addressing reviewer NTqP's comments). My concerns with respect to the computational time are also somewhat addressed.
> >
> > Regarding the reviewer hy33's comments regarding the difference between "further training" and the "two-stage retraining". Currently, I understood that the averaging over multiple seeds in this case is useful in mitigating the randomness in training. That is the methodical difference -- a) this can be more elaborated and b) conceptual differences might be less pronounced but how is this study unique --for which you can just point to all the relevant sections?

---

> > > ### Author Response · Authors · 2025-08-05
> > >
> > > Dear Reviewer 3d7B,
> > >
> > > Thank you very much for your thoughtful response to our rebuttal and for the constructive suggestions.
> > >
> > > Regarding a discussion summarizing the key points of the paper, we are happy to add one, probably as an expansion of the current Conclusion section (using the additional page allowed in the camera-ready version). We are thinking of including the following points in this discussion. Please let us know if you have additional suggestions.
> > > - Highlighting the FiMO setting and reframing the problem as one of sensitivity of the final model
> > > - Proposing further training with refinements as a gold standard for the FiMO setting
> > > - Showing theoretically that existing gradient-based TDA methods approximate further training
> > > - Showing numerically how first-order-like methods and influence-function methods approximate further training differently
> > > - Commenting on how the last point differs from findings in the literature (first-order vs. influence function methods, that the good performance of some existing methods might be due to their use of an ensemble of trained-from-scratch models or intermediate checkpoints, which are not available in the FiMO setting)
> > >
> > > Regarding Reviewer hy33's comment about "further training" vs. "two-stage retraining":
> > > - a) We can elaborate on this in the main text by bringing the relevant sentences in Appendix C into the main text.
> > > - b) We do not understand unfortunately what you mean by "how is this study unique --- for which you can just point to all the relevant sections?" What are you asking us to do?

---

> ### Comment · Reviewer_3d7B · 2025-08-05
>
> Thank you for the responses, I am in agreement with points that you have shared and believe that this will highlight the value of this work further.  Regarding the "further training" vs "two-stage retraining", I intended to suggest that the detailing in comparison should not just be at a conceptual level i.e. the difference in the methods (averaging) and it should also be highlighted that your overall contributions are still novel findings. It was more of a suggestion to frame the comparison in a broader way.
>
> Given the overall state of the paper and the. discussions, I am inclined towards acceptance and  am raising my score accordingly.

---

> > ### Author Response · Authors · 2025-08-06
> >
> > Thank you very much for raising your score and for your continued feedback!
> >
> > Regarding "further training" vs. "two-stage retraining", we were aiming to answer Reviewer hy33's question directly but agree that the comparison to Bae et al. (2022) should be framed in a broader way. Thank you for the reminder to do so.

---

### Official Review · Reviewer_2NEN · 2025-07-03

**Clarity:** 3
**Significance:** 4
**Originality:** 3
**Rating:** 5
**Confidence:** 2

**Summary:**

This paper focuses on training data attribution methods specifically in FiMO setting, where only the final model is available and neither the model checkpoints nor the training algorithm is accessible to the attribution method. They first define a taxonomy of TDA methods based on their access to different resources of the model. Then they define further training as the gold standard in TDA. This method works by training an already-trained model with different variations of the dataset generated by the leave-one-out technique. Authors further show that various attribution techniques, in fact, approximates the further training method. Experiments that were run on tabular, image, and text datasets show first-order gradient-based techniques approximate further training well, but decay over extended further training.

**Questions:**

* What's the dependency on the further training algorithm? What if the employed learning is really detrimental to the model?
* What would be the right amount of further training? Can there be a generalized guidance based on empirical evidence?

**Ethical Concerns:**

["NO or VERY MINOR ethics concerns only"]

**Final Justification:**

I really liked this paper for the gold standard framework they proposed, their discussion on non-convergence and stochasticity in NN training, and the unifying view of other methods. The authors have responded to my questions and addressed the points about time complexity of these methods and input saturation. They haven't provided an answer on the practical case of correlated and duplicate datasets, though. Some of the points raised by other reviewers about the positioning of the paper also made me reconsider my positivity towards the paper.

**Limitations:**

Authors do a great job in discussing the limitations in cost, convexity, stochasticity.

**Paper Formatting Concerns:**

None noted.

**Quality:**

4

**Strengths And Weaknesses:**

## Overall Review
Explicit definition of FiMO setting and the proposal of further training as a gold standard are timely and sound contributions. Authors connecting multiple well-known TDA techniques is a valuable theoretical framework and highlights the significance of this work in the model interpretability domain.

## Strengths
* The explicit formalization of the FiMO setting and the proposal of "further training" as gold standard are novel contributions.
* Acknowledging and addressing challenges like non-convergence and stochasticity in NN training enhances the practical relevance of the gold standard.
* Theoretical unification of first-order methods like Grad-Dot and influence function methods like CG, LiSSA, TRAK, and DataInf is very valuable.
* Reasonable empirical evaluations.
* Authors do a great job in challenging many assumptions subtly made in other papers such as "we do not assume that the empirical risk $R(D', θ)$ is convex, nor do we assume that the given parameters $θ_f$ are a stationary point of $R(D, θ)$ over the full training set D."
* The paper has a solid theoretical foundation.

## Weaknesses
* While somewhat orthogonal to the idea, authors fail to properly address dataset issues such as duplicate or high-correlated data especially for $D'$ with LOO methodology.
* While the paper explicitly mentions the cost, this method is extremely expensive nonetheless -- making it impractical for most users. This is the main reason why I am not voting Strong Accept.
* Lastly, although it is an artifact of the FiMO setting, sensitivity *after* having trained the model might not fully capture the real impact of the training example as the model may have saturated for that input already.

## Clarity
The paper is well-written and structured. The concepts, notations, and proofs are clear.

## Originality
FiMO framework formulation and further training with expectations over randomness are original contributions.

---

> ### Author Rebuttal · Authors · 2025-07-30
>
> Thank you very much for your positive review (bordering on Strong Accept) and recognition of our contributions! Below we respond to your concerns and questions.
>
> > While the paper explicitly mentions the cost, this method is extremely expensive nonetheless -- making it impractical for most users.
>
> To be clear, we are proposing further training as a gold standard and as such, we did not intend it to be computationally efficient, nor are we proposing it as a practical TDA method. The suggestions given in Appendix F (and work cited by Reviewer hy33) could ease implementation of further training in future work.
>
> > although it is an artifact of the FiMO setting, sensitivity after having trained the model might not fully capture the real impact of the training example as the model may have saturated for that input already.
>
> We agree that saturation is possible. Nevertheless, we did find in Appendix E.2 that further training influence scores can be useful for detecting mislabelled examples and giving insight into what the model has learned.
>
> > What's the dependency on the further training algorithm? What if the employed learning is really detrimental to the model?
>
> We addressed this question partially in Appendix E.1, Figure 4, where we used Adam as the optimizer instead of SGD, with qualitatively similar results.
>
> > What would be the right amount of further training? Can there be a generalized guidance based on empirical evidence?
>
> Indeed, we pose this as an open question in line 380 and elaborate on it in Appendix F, but as such is difficult to provide a concrete answer.

---

> > ### Comment · Reviewer_2NEN · 2025-08-03
> >
> > I appreciate the responses from the authors to my questions. I still like the paper's potential but have a more negative outlook than I previously had based on these two points:
> > * My first point about duplicate or correlated data (which is a fact of life in almost all real datasets).
> > * I wasn't aware of [4] from Reviewer hy33. It takes away from novelty. Also the main comment from Reviewer NTqP.
> >
> > I will keep my score the same, though.

---

> > > ### Author Response · Authors · 2025-08-03
> > >
> > > Dear Reviewer 2NEN,
> > >
> > > Thank you very much for reading our rebuttal and for your confirmation.
> > >
> > > Regarding the main comment from Reviewer NTqP, please see the follow-up response to it that we just posted. In short, we believe we are indeed reframing the problem in the FiMO setting (the "explicit formalization" that you noted).
> > >
> > > Regarding [4] from Reviewer hy33, please see our rebuttal to hy33 where we place [4] in the same category as [1] from hy33, i.e., they reverse the roles of training and test data, but do not address the FiMO setting or the question of an alternative gold standard for FiMO. Is there something specific about [4] (as opposed to [1]) that concerns you?

---

### Decision · Program_Chairs · 2025-09-17

**Decision:**

Accept (poster)

**Comment:**

The manuscript has been reviewed by four reviewers.

The authors provided a rebuttal. Eventually, 2 reviewers gave acceptance and 2 gave borderlines, indicating that no reviewers are willing to (even weakly) reject the manuscript.

Despite the minor flaws, the reviewers found the manuscript novel, providing a unifying view of gradient-based methods, and presenting extensive analysis to support the theoretical studies.

There is no ground to overturn the consensus. The AC recommends acceptance of the manuscript.

Please, still, account for the comments in the final version.